# Structural basis for dimerization of a paramyxovirus polymerase complex

Jin Xie [1,6], Mohamed Ouizougun-Oubari [2,6], Li Wang[3,6], Guanglei Zhai[1], Daitze Wu[3], Zhaohu Lin[1], Manfu Wang[4], Barbara Ludeke[2], Xiaodong Yan[4], Tobias Nilsson [5], Lu Gao [3] ✉, Xinyi Huang [1] ✉, Rachel Fearns [2] ✉ & Shuai Chen [1] ✉

The transcription and replication processes of non-segmented, negative-strand RNA viruses (nsNSVs) are catalyzed by a multi-functional polymerase complex composed of the large protein (L) and a cofactor protein, such as phosphoprotein (P). Previous studies have shown that the nsNSV polymerase can adopt a dimeric form, however, the structure of the dimer and its function are poorly understood. Here we determine a 2.7 Å cryo-EM structure of human parainfluenza virus type 3 (hPIV3) L–P complex with the connector domain (CD′) of a second L built, while reconstruction of the rest of the second L–P obtains a low-resolution map of the ring-like L core region. This study reveals detailed atomic features of nsNSV polymerase active site and distinct conformation of hPIV3 L with a unique β-strand latch. Furthermore, we report the structural basis of L–L dimerization, with CD′ located at the putative template entry of the adjoining L. Disruption of the L–L interface causes a defect in RNA replication that can be overcome by complementation, demonstrating that L dimerization is necessary for hPIV3 genome replication. These findings provide further insight into how nsNSV polymerases perform their functions, and suggest a new avenue for rational drug design.

The non-segmented negative-strand RNA viruses (nsNSVs) include numerous human pathogens, such as respiratory syncytial virus (RSV), human parainfluenza viruses (hPIVs), rabies virus (RABV), Ebola virus (EBOV) and Nipah virus (NiV)[1]. HPIV3, one of four hPIV subtypes, is a common cause of severe respiratory infections in infants and children[2,3]. The RNA polymerase complex of hPIV3, like other nsNSV members, consists of the large protein (L) and a co-factor protein, the phosphoprotein (P; or VP35 in the case of the filoviruses)[4] (Fig. 1a). L harbors five domains including the RNA dependent RNA polymerase (RdRp) domain, polyribonucleotidyltransferase (PRNTase) domain,

connector domain (CD), methyltransferase (MTase) domain and the C-terminal domain (CTD), and possesses enzymatic activities required for RNA synthesis, cap addition and cap methylation. The P protein consists of an N-terminal domain (NTD), central oligomerization domain (OD) and C-terminal X domain (XD). The P protein interacts with both L and nucleoprotein (N), allowing the polymerase to associate with the ribonucleoprotein (RNP) template and to bring soluble N proteins for encapsidation of newly synthesized replicative RNA[5–8]. The past few years have witnessed progress in the structural studies of the L–P polymerases from several nsNSVs, including the

[1]Roche Pharma Research and Early Development, Lead Discovery, Roche Innovation Center Shanghai, 201203 Shanghai, China. [2]Department of Virology, Immunology & Microbiology, Boston University Chobanian & Avedisian School of Medicine, Boston, MA 02118, USA. [3]Roche Pharma Research and Early Development, Infectious Diseases, Roche Innovation Center Shanghai, 201203 Shanghai, China. [4]Wuxi Biortus Biosciences Co. Ltd., 214437 Jiangyin, Jiangsu, China. [5]Roche Pharma Research and Early Development, Infectious Diseases, Roche Innovation Center Basel, Basel 4070, Switzerland. [6]These authors contributed equally: Jin Xie, Mohamed Ouizougun-Oubari, Li Wang. ✉e-mail: goodlucksept2016@yahoo.com; xinyi.huang@roche.com; rfearns@bu.edu; shuai.chen.sc1@roche.com

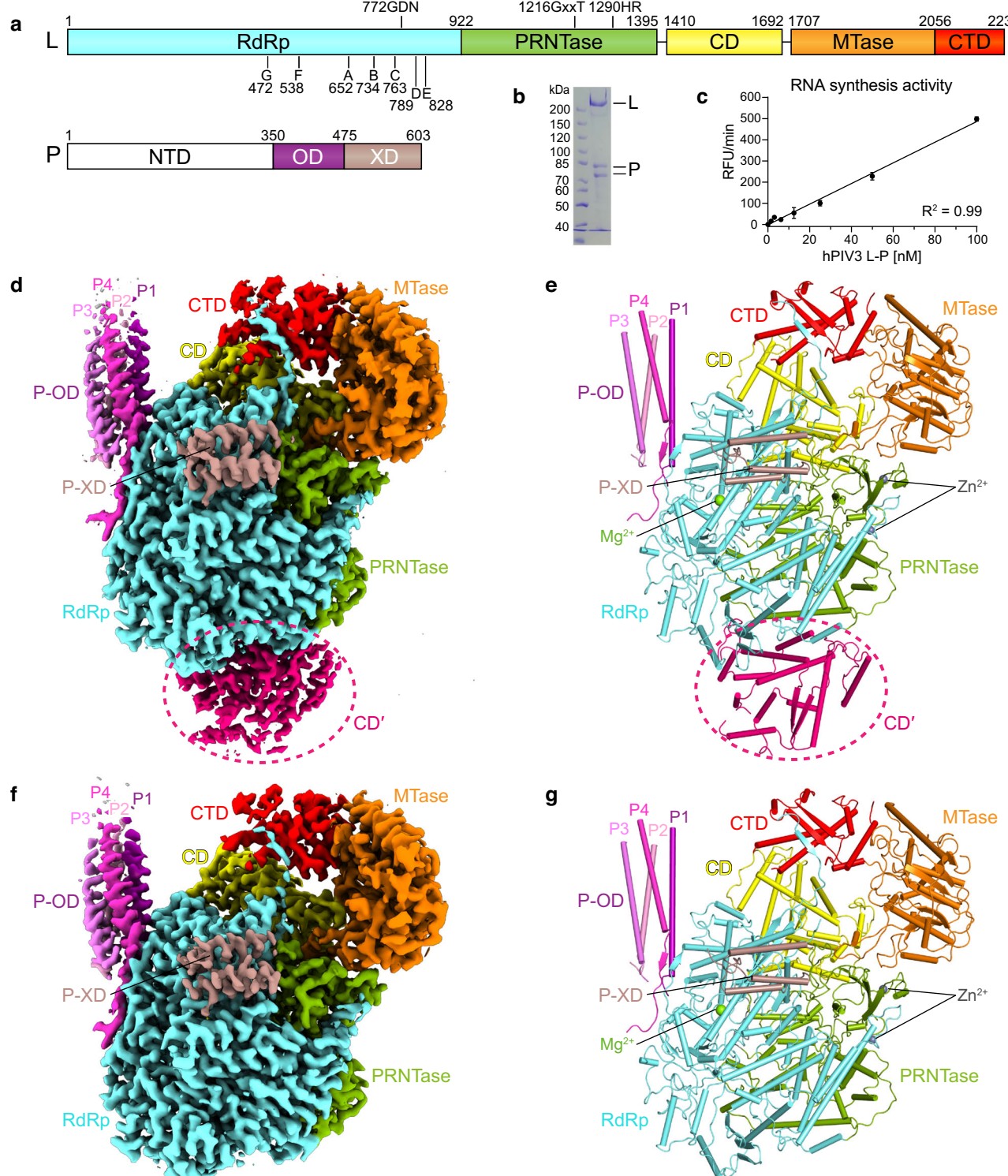

**Fig. 1 | Overall structure of the hPIV3 L−P complex. a** Domain organization of L and P proteins. RdRp, aquamarine; PRNTase, green; CD, yellow; MTase, orange; CTD, red; P-NTD, white; P-OD, purple; P-XD, dark salmon. The conserved motifs and residues required for functions are indicated. **b** SDS-PAGE analysis of the purified L−P complex. L protein and P protein have a molecular weight of 257 and 68 kilodaltons (kDa), respectively. The second P band could represent a truncated product or alternative phosphorylated state. **c** RNA synthesis activity as determined in a fluorescence-based primer extension assay. The bars show the mean and standard deviation for two independent experiments. **d**, **e** The cryo-EM map (**d**) and structure (**e**) of the hPIV3 L−P bound with the CD domain (CD′) of the second L. Domains are colored as depicted in (**a**) except for CD′ in hot pink and four copies of P-OD domains in purple (P1), light pink (P2), violet (P3) and light magenta (P4), respectively. CD′ is surrounded by a dotted circle. The magnesium ion at the RdRp active site and zinc ions in the PRNTase domain are shown as light green and gray spheres, respectively. **f**, **g** The cryo-EM map (**f**) and structure (**g**) of monomeric hPIV3 L−P. Source data are provided as a Source Data file.

rhabdoviruses, vesicular stomatitis virus (VSV)[9,10] and RABV[11], the pneumoviruses, RSV[12–16] and human metapneumovirus (hMPV)[17], the paramyxoviruses, parainfluenza virus 5 (PIV5)[6] and Newcastle disease virus (NDV)[18], and the L-VP35 complex from the filovirus, EBOV[19,20]. However, many aspects of the RNA transcription and replication mechanisms of nsNSV polymerases remain poorly understood. In particular, the nsNSV polymerase has been reported to form a functional dimer or oligomer[21–24], but the dimer structure has only been detected in low-resolution negative-stain electron microscopy (EM)[10,11,25,26]. Here, we report the cryo-electron microscopy (cryo-EM) structures of hPIV3 L–P polymerase in monomeric and incomplete dimeric forms. Only a CD domain (CD′) of the second L protein can be reconstructed to a high resolution, which is possibly due to the structural heterogeneity of the second L–P. This study reveals details of the RdRp active site and L–P binding as well as the structural conservation. We identify a unique β-strand latch of hPIV3 L that is essential for polymerase activity and provide structural correlation between the conformations of priming and intrusion loops and the rearrangements of CD, MTase and CTD domains of L. Moreover, the structural basis of L–L dimerization is presented, and functional studies reveal that the polymerase dimer is required for hPIV3 genome replication.

## Results and discussion

### Overall structure of hPIV3 L–P polymerase complex

Full-length human PIV3 L and P proteins (Fig. 1a) were co-expressed in *Spodoptera frugiperda* 21 (Sf21) cells. The RdRp activity of the purified L–P complex (Fig. 1b) was verified by a fluorescence-based primer extension assay[27] (Fig. 1c). We employed single-particle cryo-EM to solve the hPIV3 L–P complex structure at a high resolution up to 2.7 Å (class 1), allowing us to build a detailed atomic model (Fig. 1d, e, Supplementary Fig. 1, and Supplementary Table 1). All five domains of L protein and four copies of OD domain and single XD domain of P protein were built into the density map. The overall structural architecture of hPIV3 L–P is similar to that of PIV5[6] and NDV L–P[18], with corresponding sequence identities of 28% and 25%, respectively for L and both <10% for P. Interestingly, a large extra blob of electron density, not reported in previous L–P structures, was unambiguously observed near the intact L protein and was successfully assigned as the CD domain (named CD′) of a second L protein (Fig. 1d, e and Supplementary Fig. 1). In addition to this L–P structure with one L–P copy bound with CD′ of the second copy, another main class (class 2) of particles lacking density for the second CD domain was reconstructed to a 3.3 Å resolution structure representing the monomeric L–P complex (Fig. 1f, g and Supplementary Fig. 1). Since these two structures show nearly the same arrangement except for the second CD domain, we used the 2.7 Å structure for further analysis.

To see if more density of the second L–P copy could be observed, we attempted to reconstruct L–P copy 2 in class 1 by subtraction of the "complete" L–P copy. Interestingly, 2D classification yielded classes with an apparent ring-like RdRp-PRNTase core of L instead of featured CD′ (Supplementary Fig. 2a), which may be caused by the new alignments with the larger RdRp-PRNTase compared to that of CD′. Indeed, after 3D classification with good CD′ alignments derived from the refinement of class 1, a map with clear structural features of CD′ was obtained. Furthermore, we observed a large, smeared density near the density of CD′, also indicating the potential missing parts of the second copy of L–P (Supplementary Fig. 2b). Focused refinement of the CD′ part obtained a 4.5 Å map. It only differs from the original CD′ map of class 1 in the visible density of a loop and a little more density extended at the N- and C-termini of CD′, reflecting the flexibility flanking the CD′ with long loops in the second copy of L–P (Supplementary Fig. 2c). For other parts, reconstruction of the particles with clear 2D features mentioned above obtained a 7.0 Å low-resolution map of potential RdRp-PRNTase of the second L–P. This map could overall be overlaid

with the RdRp-PRNTase map and structure of class 1 and has the typical holes corresponding to the proposed template entry, template exit and RNA tunnel (Supplementary Fig. 2d). Notably, the relative orientation of the RdRp-PRNTase and CD domains of the second L could not be determined. Thus, these reconstruction results further illustrated the existence of a second L–P copy and its structural heterogeneity.

### RdRp active site of hPIV3 L

Analysis of the 2.7 Å class 1 L–P structure showed that, consistent with previously determined RNA polymerase structures[6,9,12,28–31] (Supplementary Fig. 3), the RdRp domain of hPIV3 L–P complex folds into the canonical right-hand fingers-palm-thumb subdomains, while the catalytic active site is composed of seven conserved motifs (A–G) (Fig. 2a, b and Supplementary Fig. 4). Motifs A–E are located in the palm subdomain, while motifs F and G are located in the fingers subdomain. To further understand the RdRp active site of nsNSVs, we compared the hPIV3 RdRp structure to the RNA duplex/$Mg^{2+}$-bound RdRp structures of influenza B virus (FluB)[30] and severe acute respiratory syndrome coronavirus 2 (SARS-CoV-2)[31], as representatives of segmented negative-strand RNA viruses and positive-strand RNA viruses, respectively. The hPIV3 L RdRp domain shows similar structural architecture to that of FluB and SARS-CoV-2 (Supplementary Fig. 3). Furthermore, motifs A–G could be well overlaid, and the proposed catalytic residues 772-GDN-774 of hPIV3 RdRp could be also superimposed with those of FluB (443-SDD-445) and SARS-CoV-2 (759-SDD-761) (Fig. 2c, d). One magnesium ion at the catalytic center could be built in our hPIV3 structure based on the well-resolved electron density (Fig. 2a–d and Supplementary Fig. 1g), which has not been reported in previous nsNSV L structures. The presumed catalytic $Mg^{2+}$ is located at the similar position as one of the two $Mg^{2+}$ present in FluB and SARS-CoV-2 structures and coordinated by the side chain oxygen of the catalytic residue Asp773 and the main chain oxygen of motif A residue Leu664 (Fig. 2c, d). Similar to FluB[30] and SARS-CoV-2[31], hPIV3 L appears to possess the conserved motif F residues Arg552 and Lys543/Phe554 to stabilize the incoming nucleotide and the template strand RNA at the +1 site, respectively. In addition, hPIV3 also has a conserved positively charged residue, Lys475 within motif G that could function to direct the template strand RNA into the active site[30,31] (Fig. 2c, d and Supplementary Fig. 4). During our manuscript preparation, the EBOV L-VP35 and RSV L–P structures in complex with a 10-nt single-strand template RNA were published[15,20]. Interestingly, Phe554/Lys475/Lys543 of hPIV3 L identified here are consistent with Phe563/Arg485/- of EBOV L and Phe629/Lys540/Lys619 of RSV L among the residues involved in interactions with the template strand RNA (Supplementary Fig. 5). In addition, our results also facilitate the understanding of the catalytic $Mg^{2+}$, the +1 site and interactions with template-product RNA duplex of nsNSV RdRp active site as described above.

### HPIV3 L adopts a distinct conformation

The RdRp is juxtaposed with the multifunctional PRNTase domain, which is involved in RNA synthesis initiation, capping and elongation[32–37] to form the rigid core of L, appended with the flexible CD, MTase and CTD domains[6,26] (Supplementary Fig. 1f). Structural comparison reveals that some features within the PRNTase and the structural arrangement of the CD, MTase and CTD of hPIV3 are different from other reported nsNSV structures[6,9–13,17] (Fig. 2e, f and Supplementary Figs. 6, 7). Moreover, the conformations of priming and intrusion loops appear to be coupled with the rearrangements of the flexible C-terminal appendages of L. In the VSV and RABV structures, the priming loop that is thought to facilitate synthesis initiation reaches towards the RdRp active site, likely representing a pre-initiation state[9–11]. In contrast, the hPIV3 priming loop (Leu1210–Ser1234) retracts considerably from the RNA cavity. Instead of the priming loop, an intrusion loop (Pro1281–Ser1305), containing the catalytic HR motif for capping, partially projects out into the

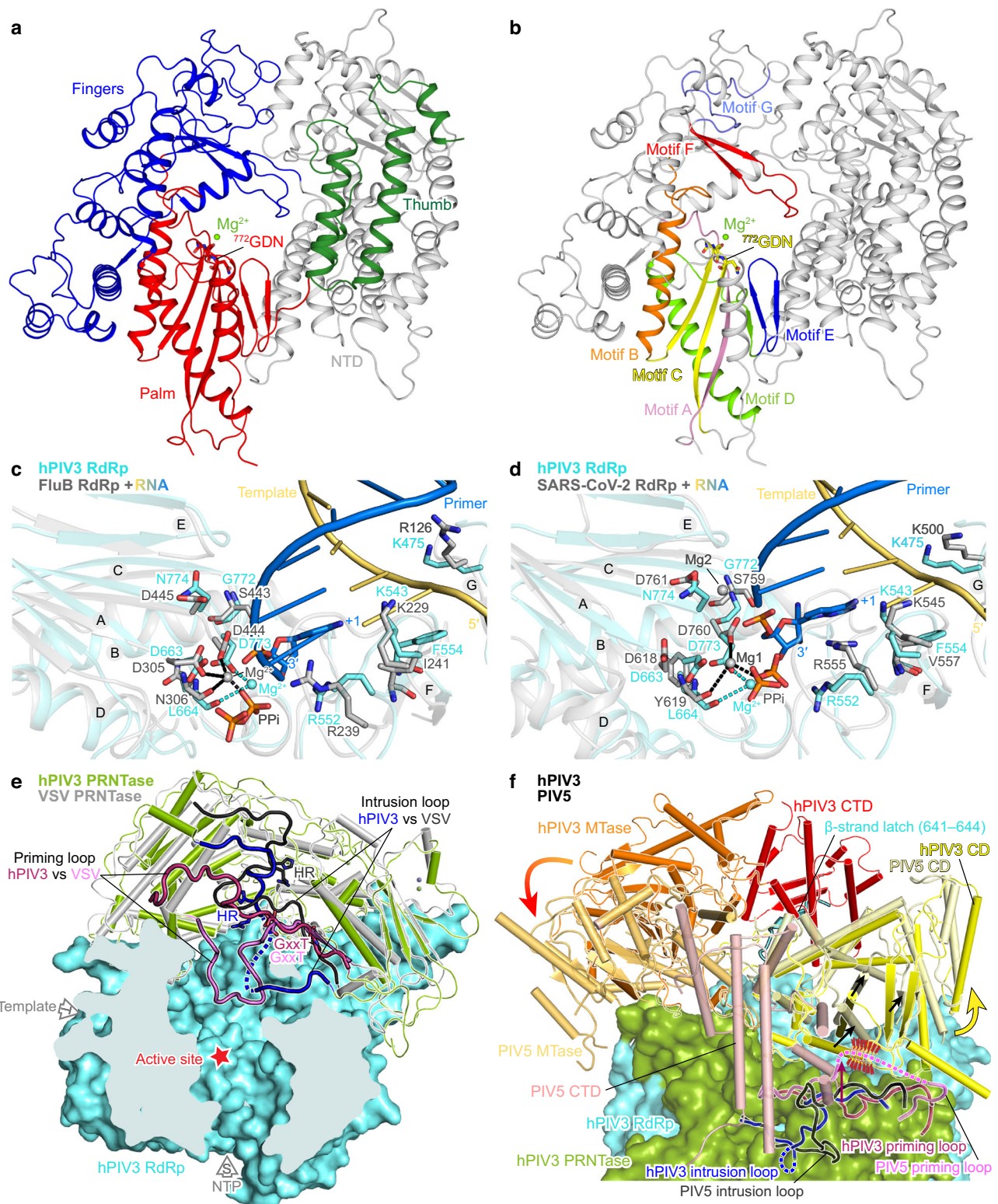

central tunnel, a feature also observed in PIV5 L[6] (Fig. 2e and Supplementary Fig. 6a–d). It is to be noted that, the HR motif on the intrusion loop is directed towards the RNA cavity in the hPIV3 structure, but situated away from the cavity in the structures of hPIV5 and other nsNSVs (Fig. 2e and Supplementary Fig. 6). Accompanying the different conformations of priming and intrusion loops, there are corresponding rearrangements of CD, MTase and CTD (Fig. 2f and Supplementary Fig. 7). Compared to VSV and RABV structures, the retracted priming loop of hPIV3 L props against CD, leading to a slight

shift of CD and subsequent movement of MTase-CTD (Supplementary Fig. 7a–c). In contrast, the priming loop of PIV5 retracts further than that of hPIV3, resulting in a shift of PIV5 CD away from the cavity. The adjacent MTase-CTD module of PIV5 subsequently undergoes a rotation, positioning the MTase active site closer to the capping site[6] (Fig. 2f and Supplementary Fig. 7d). In the RSV and hMPV structures that adopt a non-initiation state[12,13,17,28], the priming and intrusion loops are fully retracted, possibly leading to a flip of CD with more significant movement of MTase-CTD to expose the product exit channel

**Fig. 2 | Structural architecture of the hPIV3 L protein. a** The RdRp domain is shown in ribbons, with the fingers subdomain in blue, the palm subdomain in red, the thumb subdomain in dark green, and the N-terminal region (NTD) in gray. The catalytic residues 772-GDN-774 and the magnesium ion (Mg$^{2+}$) at the active site are shown as sticks and sphere, respectively. **b** Motifs A–G of the RdRp domain are highlighted in rainbow colors with the same view as in (**a**). **c, d** RdRp active site. The RdRp domain of hPIV3 is superimposed on that of RNA duplex/Mg$^{2+}$-bound RdRp structures of FluB (PDB 6QCX) (**c**) and SARS-CoV-2 (PDB 7BV2) (**d**). Motifs A–G are labeled as A–G. The hPIV3 structure is colored in aquamarine, while FluB and SARS-CoV-2 structures are colored in gray except for the template and primer (product) strand RNAs in light orange and marine blue, respectively. The nucleotide at the +1 position of the primer strand and the pyrophosphate (PPi) are shown as sticks. The catalytic residues and critical conserved residues that interact with RNA are also

shown as sticks. **e** The PRNTase domain (green) of hPIV3 L. A superposition of hPIV3 L with VSV L (PDB 6U1X) (gray) highlights the conformational differences of the putative priming loops and intrusion loops. The priming loops of hPIV3 and VSV are colored in claret and pink, respectively, while intrusion loops are in blue and black, respectively. The disordered region of hPIV3 intrusion loop is shown as a dotted line. The conserved GxxT motif and HR motif are labeled. The hPIV3 RdRp domain is shown as aquamarine surface. The red star indicates the RdRp active site. **f** Superposition of CD-MTase-CTD of hPIV3 (yellow-orange-red) and PIV5 (PDB 6V85) (similar colors). RdRp-PRNTase of hPIV3 is shown as transparent surface. The disordered regions of the loops are shown as dotted lines. Potential steric clash between PIV5 priming loop and the CD adopting the same conformation as hPIV3 is indicated by red dashes. Movements are indicated by arrows. The β-strand latch (Phe641–Lys644) of hPIV3 L RdRp is labeled.

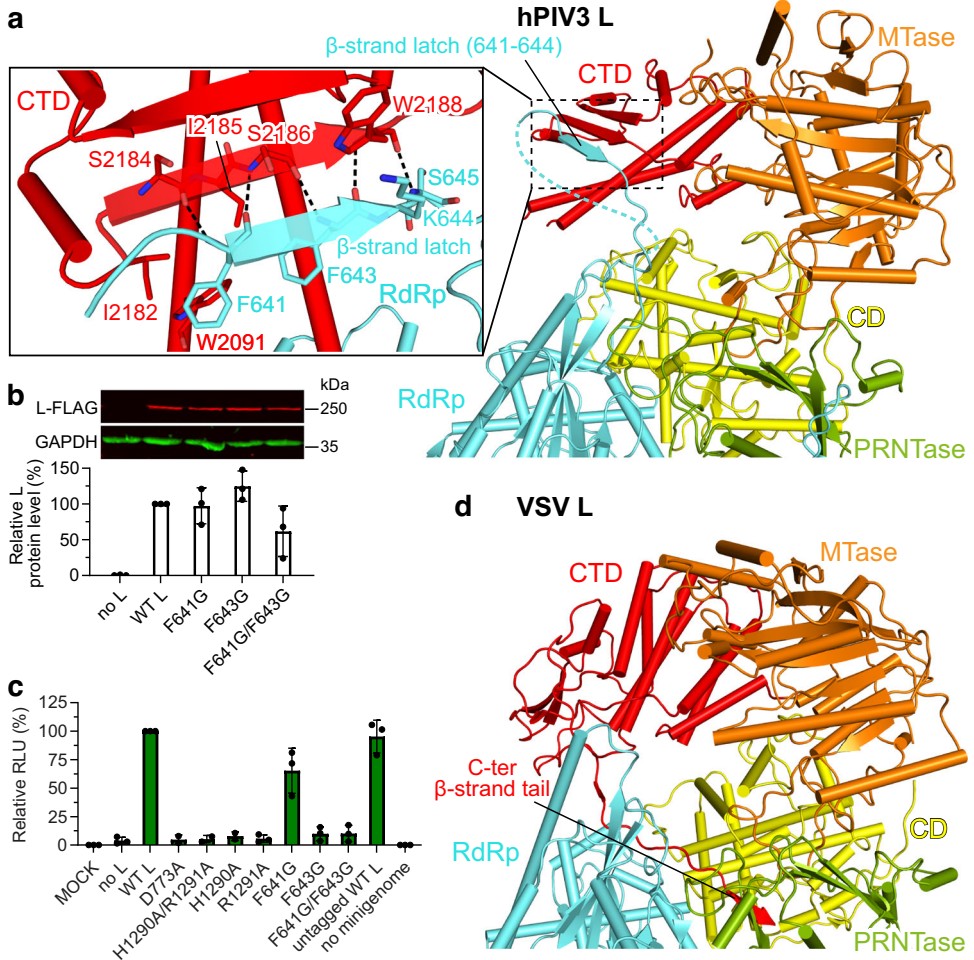

**Fig. 3 | The unique β-strand latch of hPIV3 L RdRp for transient positioning of CTD. a** The β-strand latch extended from hPIV3 L RdRp forms interactions with L CTD. The domains of hPIV3 L are colored as depicted in Fig. 1a. The β-strand latch (Phe641–Lys644) of hPIV3 L RdRp is labeled. The disordered region followed by the β-strand latch is shown as a dotted line. Magnified view of the interactions between β-strand latch and CTD of hPIV3 are shown in the left panel and the polar interactions are indicated by dashed lines. **b** Western blot to confirm the expression of FLAG-tagged variant L proteins with mutations on the β-strand latch. Representative gel and quantification of the protein levels relative to that of the FLAG-tagged wild-type (WT) L are shown in the upper and lower panels, respectively. **c** Luciferase-based minigenome assay to evaluate the polymerase activity of the β-

strand latch mutants. The mutants with substitutions on the RdRp catalytic residue Asp773 or HR motif (His1290 and Arg1291) of PRNTase domain, which would be expected to inhibit polymerase activity are used as negative controls. Luciferase levels are normalized to an internal control and then to FLAG-tagged WT L, which is set to 100%. Relative luciferase units, RLU. The bars in **b** and **c** show the mean and standard deviation for three independent experiments (the data points for each experiment are shown). **d** The C-terminal β-strand tail of VSV L CTD that may have similar function. The C-terminal β-strand tail (β-strand Ser2106–Asp2109 with the upstream loop) of VSV L CTD is labeled. Source data are provided as a Source Data file.

(Supplementary Fig. 7e, f). Consistent with this, their CD-MTase-CTD domains were not resolved in the EM maps[12,13,17]. In addition, the recently published NDV L structure appears to have similar overall arrangements of CD-MTase-CTD and priming and intrusion loops to

that of hPIV3, however the priming loop has a different contact to CD corresponding to a slightly different positioning of CD-MTase-CTD compared to hPIV3 (Supplementary Fig. 8). Although it seems likely that the different structures described represent different

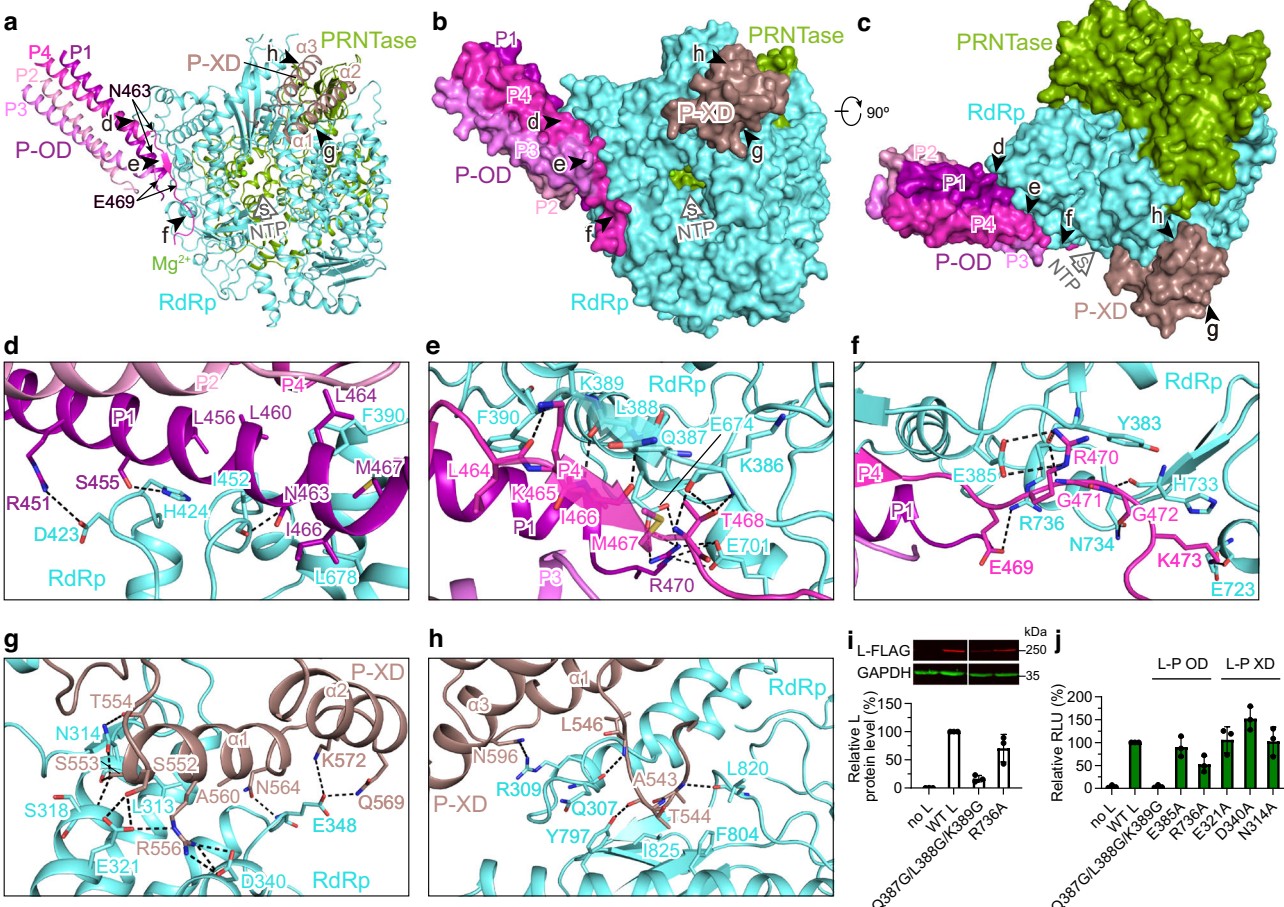

**Fig. 4 | L−P interaction of hPIV3. a** Overview of OD and XD domains of P binding to L. Five black arrows represent the views shown in (**d−h**). L and P are shown as ribbons. The region (Asn463-Glu469) that is liberated from the long helix of P4 is labeled. The putative NTP entry is indicated by a thick arrow. **b** The same view as in (**a**) with L and P shown as surface. **c** Rotated view by 90° about the horizontal axis from (**b**). **d−f** Magnified view of the interactions between L and P-OD. **g, h** Magnified view of the interactions between L and P-XD. Polar interactions are indicated by dashed lines. **i, j** Expression level (**i**) and polymerase activity (**j**) evaluation of hPIV3 L mutants with substitutions on the L−P OD interface and the L−P XD interface, as measured by Western blot analysis and a luciferase minigenome assay, respectively. The bars show the mean and standard deviation for three independent experiments (the data points for each experiment are shown). Note that (**i**) shows a single Western blot with intervening lanes excised. Source data are provided as a Source Data file.

conformational states, how these different conformations correlate to different stages of transcription versus replication remains unknown.

An unusual short β-strand extended from the hPIV3 L RdRp (referred to as the β-strand latch) aids the positioning of the distal CTD via forming a β-sheet and hydrophobic interactions (Figs. 2f, 3a and Supplementary Fig. 4). F643G and F641G/F643G substitutions on this β-strand latch nearly abolished hPIV3 polymerase activity, while F641G also had a moderate effect (Fig. 3b, c). It reveals that the β-strand latch is essential for hPIV3 polymerase function through stabilizing the flexible C-terminal appendages of L. The short β-sheet might be susceptible to separation during RNA elongation to adjust the rearrangement of C-terminal domains for RNA channel opening (Supplementary Fig. 9). Intriguingly, this structure feature has not been observed in polymerase structures of other nsNSVs, including the paramyxoviruses PIV5 and NDV (Fig. 2f and Supplementary Fig. 8a), but is only conserved in the closely related paramyxoviruses such as human PIV1 (hPIV1) and Sendai virus (SeV) (Supplementary Fig. 4). It appears that VSV L inserts its C-terminal β-strand tail of the CTD into RdRp-PRNTase for similar function of transient fixation (Fig. 3d and Supplementary Fig. 4). Coincidentally, the polymerases of some segmented negative-strand RNA viruses, such as La Crosse virus and severe fever with thrombocytopenia syndrome virus were reported to support and stabilize the C-terminal region utilizing a protruding β-hairpin strut or lariat in certain functional states[38–40].

## Unique and conserved L−P binding

Detailed L−P interactions are observed in the hPIV3 structure (Fig. 4a–h). Tetrameric OD domains of hPIV3 P constitute a long helical bundle bound to the RdRp domain of L. Each of the four P monomers (P1–P4) adopts asymmetric conformations. Two proximal subunits P1 and P4 make extensive contacts with L (Fig. 4a–f). Residues Asn463–Glu469 of P4 are liberated from the long helix, and residues Lys465–Met467 form a typical antiparallel β-sheet with Gln387–Lys389 of L (Fig. 4a, e). In addition, the neighboring residue Phe390 along with Ile452 and Leu678 of L inserts into the exposed hydrophobic core composed of P1 and P4 (Fig. 4d, e). Similar β-sheet (Supplementary Fig. 11) and hydrophobic interactions are also present in NDV, RSV and hMPV structures[12,13,17,18]. Among the three C-terminal α helices (named α1–α3 here) of hPIV3 P-XD, the α1 and its upstream loop, corresponding to the single α-helix at the C-terminus and the neighboring linker observed in RSV and hMPV L−P structures[12,13,17] (Supplementary Fig. 11), contribute to the majority of interactions between P-XD and L (Fig. 4g, h). Analysis of a panel of L mutants with substitutions in the P-OD or P-XD interacting regions using the minigenome system showed that, except for an R736A substitution, single residue substitutions had marginal effects (Fig. 4i, j), likely due to the broad network of interactions between L and P. A triple mutant Q387G/L388G/K389G that would be predicted to destabilize the β-sheet with P4-OD (Fig. 4e) reduced L protein expression to approximately 17% of

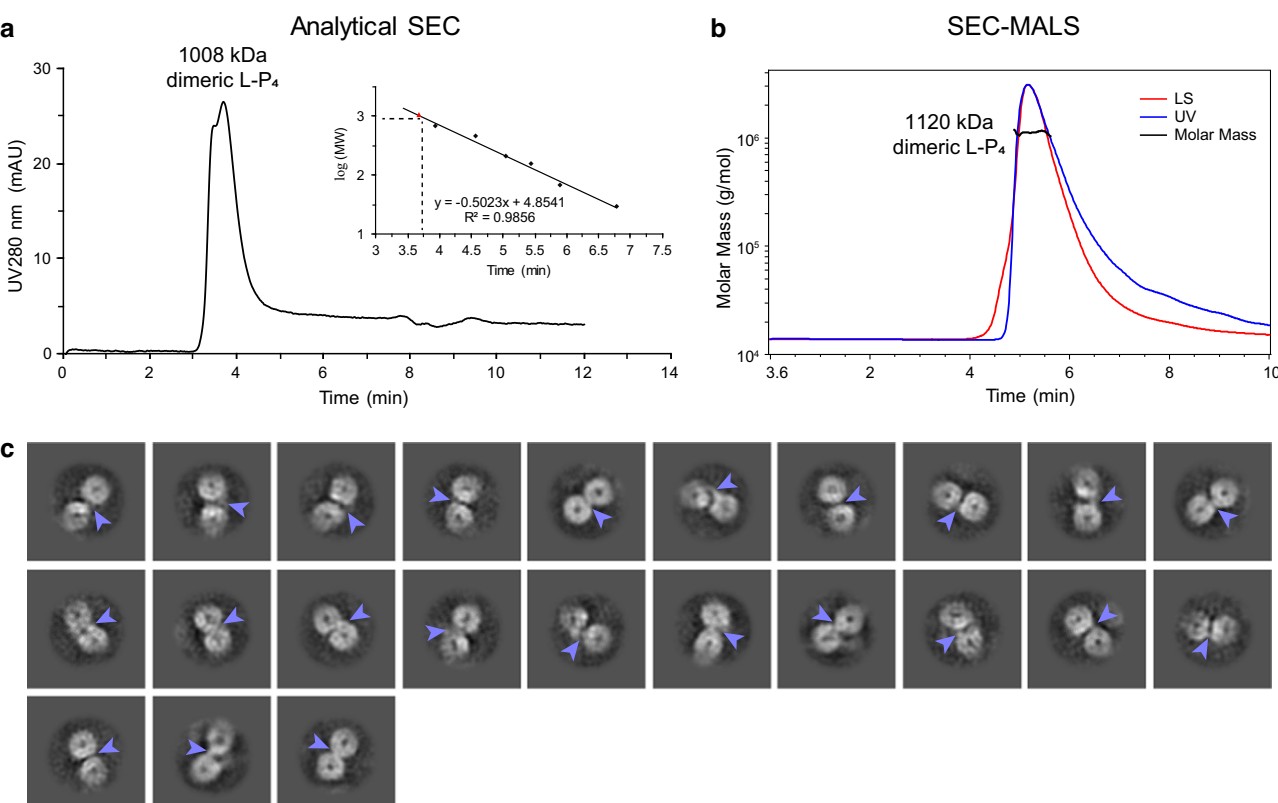

**Fig. 5 | Dimeric L–P polymerase of hPIV3. a** Molecular weight determination of hPIV3 L–P complex by analytical size-exclusion chromatography (SEC). mAU, milliabsorbance units. The theoretical molecular weight of the L–P complex (L-P₄) dimer is 1,058 kDa. **b** Molecular weight determination of hPIV3 L–P complex by size-exclusion chromatography with multi-angle light scattering (SEC-MALS). LS, light scattering; UV, ultraviolet. **c** 2D classification of the putative entire L–P dimers in single particle reconstruction by cryo-EM. Light blue arrows indicate the potential parts of other domains to bridge two L cores composed of RdRp-PRNTase domains.

wild-type levels (Fig. 4i). This is likely because P is a chaperone for L. However, this mutation had an even more pronounced effect on L polymerase activity, reducing it to background levels (Fig. 4j).

Most of the residues involved in the L–P interactions are highly conserved in the closely related paramyxoviruses such as hPIV1 and SeV, while less conserved in the distantly related paramyxoviruses PIV5 and NDV and other nsNSVs (Supplementary Figs. 4, 10). However, the docking positions of P on L and some interaction features are similar among hPIV3 and several other nsNSVs (Supplementary Fig. 11), as mentioned above, reflecting that the L–P binding mode seems to be conserved among the *Paramyxoviridae* and *Pneumoviridae* families even though P proteins are relatively diverse and the individual interacting residues are varied. The EBOV L-VP35[19] binding arrangement is also similar even though the sequence similarity between P and VP35 is very low.

### Structural definition of the L–L interaction

Genetic and biochemical evidence showed that L proteins of hPIV3[23], hPIV2[24], and SeV[22] could form dimers or oligomers, and dimeric L–P complexes of RABV[11] and VSV[10,25,26] were observed under negative-stain EM; however, the detailed structures and interactions remain unclear, hindering our understanding of how nsNSV polymerases function. Here, we obtained the structure of hPIV3 L–P with the clearly traced CD domain (CD′) of a second L protein by 3D reconstruction (Fig. 1d, e and Supplementary Fig. 1h). Reconstruction of the second L–P copy of this particle class further illustrated the existence of an intact second copy of L–P (Supplementary Fig. 2). Consistent with this, molecular weight determination of the purified L–P complexes by analytical size-exclusion chromatography (SEC) and size-exclusion chromatography with multi-angle light scattering (SEC-MALS) indicated that most of the sample contained complete L–P dimers, rather than L–P monomers

complexed with a fragment of L (Fig. 5a, b). Meanwhile, we found some different 2D classes that were discarded in previous data processing due to poor quality. After several rounds of 2D classification of these particles, we obtained some class averages of putative entire L–P dimers, with two apparent ring-like RdRp-PRNTase cores and some other parts including the potential bridging regions between two L (Fig. 5c). However, these 2D classes of dimers were not able to be reconstructed further. Based on these findings, we can conclude that the structure that we solved is part of a complete L–P dimer, and that the reason for the failure of 3D reconstruction to get a complete dimeric structure may be the structural heterogeneity due to different conformations with dynamic C-terminal CD-MTase-CTD domains of L. As noted above, the CD-MTase-CTD domains were not resolved in RSV and hMPV L–P structures although full-length L proteins were used[12–14,17], and only monomeric L–P complex structures of RABV and VSV were reconstructed although 2D classes of dimers could be observed under negative-stain EM[10,11], consistent with structural heterogeneity. In addition, we cannot exclude the possibilities of the preferred orientation problem of the particles and protein denaturation at the air-water interface preventing the reconstruction of the entire dimer.

In the hPIV3 L–P structure with CD′ of the second L, CD′ contacts the RdRp and PRNTase domains of the visible full L, with a large total buried interface of 2452 Å² composed of electrostatic and hydrophobic interactions (Fig. 6a, b). Other regions of the second L are less likely to contribute to key interface between two L in our captured polymerase "dimer", because no additional densities bound to the intact L were observed during reconstruction, and usually an interface is relatively stable. Therefore, our reported interface here should represent an L–L dimer interface, at least at this state we captured. At the interface, the charged residues

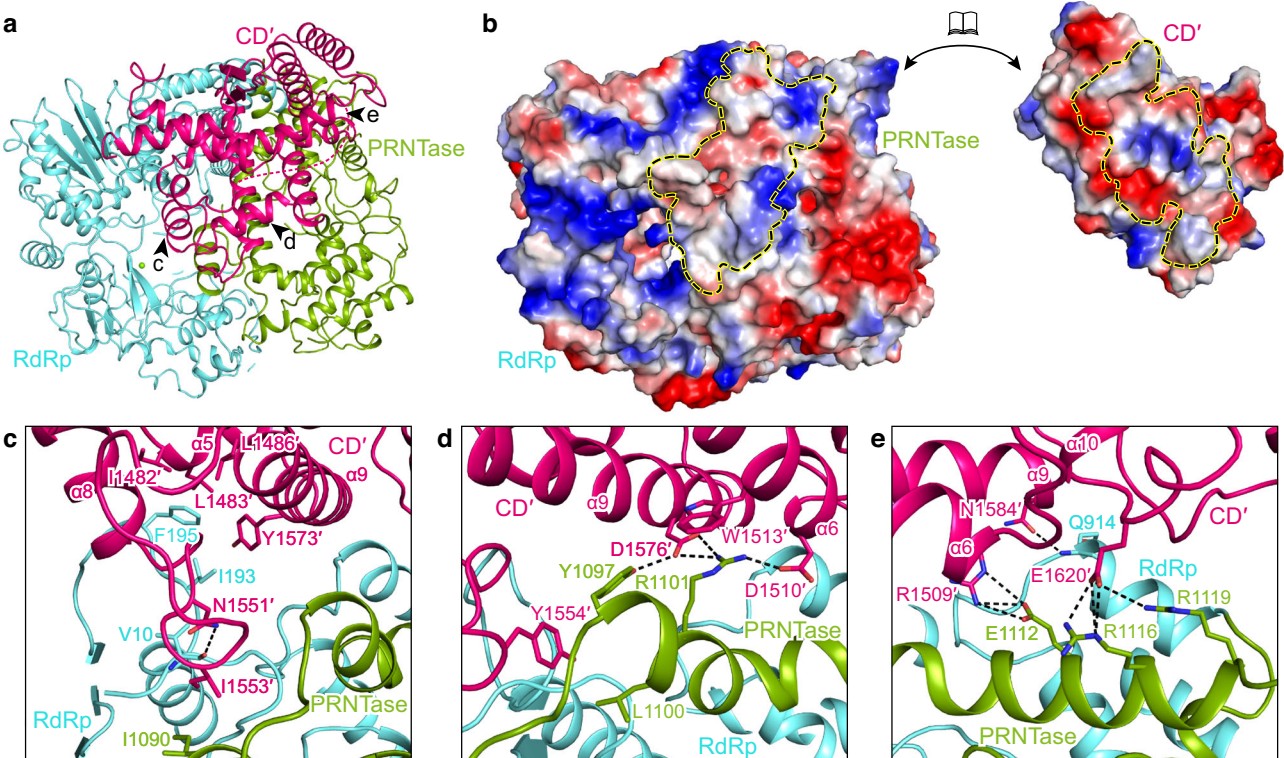

**Fig. 6 | L–L interaction. a** Overview of the CD domain (CD′) of the second L binding to the RdRp and PRNTase domains of the intact L. Three arrows represent the views shown in (**c–e**). **b** The surfaces of CD′ and RdRp-PRNTase are colored by electrostatic potential from red to blue (negative to positive, respectively). The interface of each is indicated by a dashed circle. **c–e** Magnified view of the interactions between CD′ and RdRp-PRNTase. The polar interactions are indicated by dashed lines.

Arg1509, Asp1510, Asp1576 and Glu1620 on CD′ form salt bridges with Glu1112, Arg1101, Arg1116 and Arg1119 on a long α-helix of PRNTase (Fig. 6c–e). Some hydrogen bonds and hydrophobic interactions are also formed between residues of CD′ on helices α5 and α9 along with its upstream loop and residues from both RdRp and PRNTase domains of the intact L (Fig. 6c–e). In addition, Trp1513 of CD′ forms a cation–π interaction with Arg1101 of PRNTase (Fig. 6d). CD′ has nearly the same overall structure as the CD domain of the intact L, except for the two loops that contribute to the L–L interface and the bent helix α2 (Asp1430–Asp1456) of CD′ (Supplementary Fig. 12a). Interestingly, adopting such a bent α2 conformation could cause potential steric clashes for the interface between CD and RdRp-PRNTase within one L. The structure reveals that the two CD domains can exquisitely bind to distinct interfaces on RdRp and PRNTase domains (Supplementary Fig. 12b), thus acting as both intramolecular and intermolecular connectors. Although the CD domain has no known enzymatic activity, it cannot tolerate in-frame insertions and domain exchanges between the substrains of VSV[41], consistent with it playing a significant role. Most of the residues involved in the L–L interactions are conserved among the closely related paramyxoviruses; they were less conserved in other nsNSVs, but some comparable interactions appear to exist in these nsNSV L (Supplementary Fig. 4). For example, Arg1101 and Asp1576 that form important salt bridges in hPIV3 are not conserved in RSV, hMPV, VSV and RABV, but the interaction is possibly achieved by the corresponding residue pairs Asp1163/Lys1608 (RSV), Asp1088/Lys1529 (hMPV), Asp1053/Arg1493 (VSV), Glu1067/Arg1511 (RABV) with opposite charges. In addition, the hPIV3 CD domain shares a very similar structure with that of other reported nsNSV structures, despite low sequence identities (Supplementary Fig. 12c). Therefore, we cannot exclude the conservation of L–L dimerization across paramyxoviruses and even other nsNSVs.

## Dimeric L–P polymerase in RNA replication

To assess the functional significance of L–L dimerization, mutation analysis was performed using a dicistronic minigenome assay for transcription and RNA replication. Because the multifunctional nature of the RdRp and PRNTase domains could complicate data interpretation, mutations were designed and introduced into the CD′ to disrupt the key salt bridges and cation–π interaction at the dimer interface (Figs. 6d, e and 7a, b). Among the single residue substitutions on the CD side of L–L dimer interface, W1513A and D1576A generated reduced amounts of antigenome RNA, with only minor changes in mRNA levels, while other mutants had subtle or negligible effects (Fig. 7a, b), possibly due to compensatory effects from other interactions. A triple mutant R1509A/D1510A/W1513A had a profound effect, reducing both antigenome and mRNA to almost undetectable levels (Fig. 7a, b). It should be noted that these experiments employed a replication-competent minigenome, meaning that a defect in replication would reduce the amount of available template for mRNA synthesis. Therefore, it is not possible to determine if the reduction of mRNA levels generated by the mutant L proteins was a direct or indirect effect. However, the data clearly show that the L–L interface is necessary for RNA replication. Interestingly, SeV L mutant Y1097S/G1098R/I1099V, with Tyr1097 corresponding to hPIV3 Tyr1097 on the L–L interface (Fig. 6d), was reported to show significantly reduced activity in replication but not transcription[42].

Complementation assays were performed to confirm if dimerization of L enables RNA replication. Highly deficient L mutants were analyzed alone or in combination with the L–L interface triple mutant using the minigenome assay described above. When analyzed in the absence of complementing L protein, the β-strand latch mutant (F643G) and L–P4 OD interface mutant (Q387G/L388G/K389G) were each deficient in RNA replication; transcription levels were also disrupted substantially which could be due to reduced template levels, as

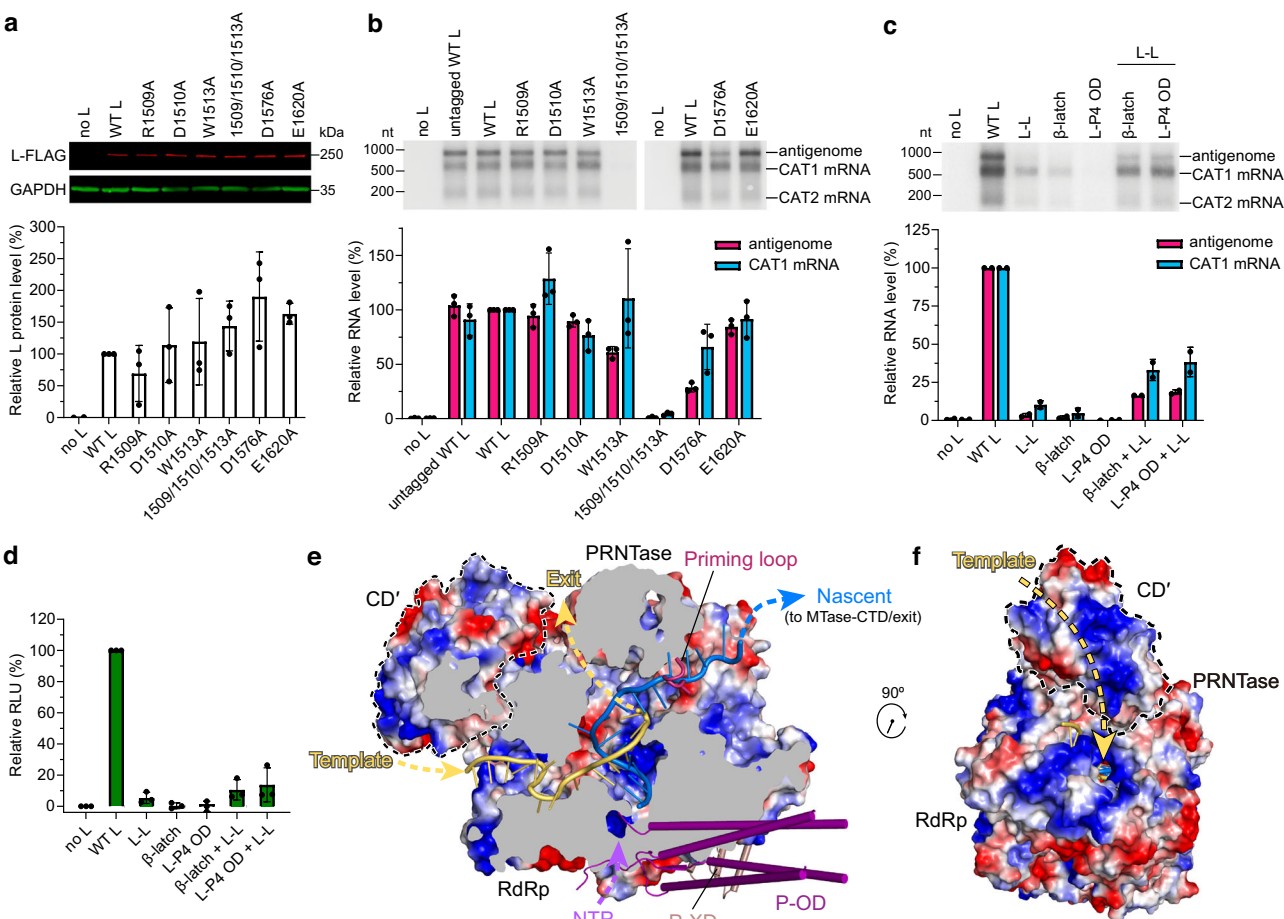

**Fig. 7 | The dimeric hPIV3 L–P plays a role in RNA replication. a** Western blot to confirm the expression of FLAG-tagged L–L interface mutants with substitutions on hPIV3 L CD domain. Representative gel and quantification of the protein levels relative to that of the FLAG-tagged wild-type (WT) L are shown in the upper and lower panels, respectively. **b** Northern blot analysis of antigenome and mRNAs generated by L–L interface mutants in a minigenome assay. The antigenome and CAT1 mRNA levels were quantified relative to that of FLAG-tagged wild-type L. The bars in (**a**, **b**) show the mean and standard deviation for three independent experiments (the data points for each experiment are shown). **c**, **d** Northern blot-based (**c**) and luciferase-based (**d**) minigenome complementation assays of the L–L interface mutant (R1509A/D1510A/W1513A), the RdRp β-strand latch mutant (F643G) and the L–P4 OD interface mutant (Q387G/L388G/K389G). The bars show

the mean and standard deviation for $n = 2$ and 3 independent experiments in (**c**) and (**d**), respectively. All the L proteins contain the C-terminal FLAG tag except for the untagged WT L with notation in (**a**) and (**b**). **e** Cut-out view of L RdRp and PRNTase domains bound with CD′ showing electrostatic surfaces and paths followed by the template (light orange) and nascent (marine blue) RNA strands modeled from FluB RdRp structure (PDB 6QCX). The putative NTP entry, template RNA entry and exit, and nascent RNA exit tunnels are indicated by dashed arrows. Electrostatic potential from negative to positive is indicated by red to blue, respectively. The part that affects the nascent RNA exit in the priming loop is omitted from the electrostatic surfaces and shown as ribbons. **f** Rotated view by approximately 90° about the z-axis. Source data are provided as a Source Data file.

noted above (Fig. 7c). Importantly, the combination of either mutant with the L–L mutant provided a clear rescue of polymerase activity (Fig. 7c). Similar results were obtained by analyzing luciferase expression from a luciferase-expressing minigenome assay (Fig. 7d). The fact that the L–P4 OD mutant is poorly expressed (Fig. 4i) suggests that it complements highly efficiently. These results show that the hPIV3 L protein adopts a dimeric form to engage in RNA replication.

Recently, structures of both symmetric and asymmetric RNA polymerase dimers were reported for influenza viruses, and these dimers are also involved in RNA replication[28,43–45]. Here, the hPIV3 L–P dimer we captured appears to be asymmetric, otherwise serious steric clashes would be introduced between two L–P monomers (Supplementary Fig. 13). As mentioned above, the fact that the rest of the second L–P was not visible in our reconstructed structure with CD′, and further separate reconstruction of the second L–P observed more regions, is possibly due to the heterogeneity with dramatic movement flanking CD′. Interestingly, CD′ is positioned adjacent to the putative template entry of the neighboring L and has a positively charged inner surface connected to the RNA tunnel, which suggests CD′ may provide

a channel to transport RNA from polymerase into the adjoining polymerase (Fig. 7e, f). The basic residues that constitute the positively charged surface of CD′ are conserved in general across different nsNSVs, and comparable surface is shown on L CD domain of the reported structures of other nsNSVs, such as NDV and VSV (Supplementary Figs. 4, 12d). Coincidentally, a similar basic path connecting the product exit channel of the replicating polymerase to the template entry on the encapsidating polymerase was also observed in the asymmetric influenza polymerase dimer that facilitates RNA encapsidation[44]. In the nsNSVs, encapsidation involves delivery of soluble N protein (N⁰) to the growing RNA chain, with P protein serving as a chaperone for this process[46]. The fact that the L–P4 OD mutant was able to complement the L–L mutant in replication, coupled with the previous finding that the SeV L mutant Y1097S/G1098R/I1099V was defective in a single round of replication (as opposed to multi-cycle replication)[42], is consistent with the hypothesis that hPIV3 L–P dimer enables encapsidation. Based on these findings, and by analogy with the influenza polymerase dimer, we propose a dimeric L–P polymerase model for RNA replication, in which one polymerase engages in RNA

synthesis and the nascent RNA product travels along the positively charged surface of the CD domain to the adjoining polymerase (Supplementary Fig. 14). However, the detailed mechanism of RNA replication and encapsidation in nsNSVs is still poorly understood; the structure and functional studies described here present a starting point for more detailed characterization of the nsNSV replicase.

Our findings presented here provide further insight into the polymerase of nsNSVs by revealing an L–L dimer interface and demonstrating a role for the dimeric hPIV3 L–P in RNA replication. These findings shed new light into our understanding of the structure-function properties of nsNSV polymerases and provide a more complete picture of the different conformations that the polymerase can adopt. Further, targeting the polymerase dimer interface with obvious pockets provides a potential novel avenue for antiviral drug design.

## Methods

### Expression and purification of hPIV3 L–P complex

The codon-optimized sequences for human PIV3 L (GenBank accession code: NP_067153.2) and P (GenBank accession code: NP_067149.1) proteins were chemically synthesized and cloned downstream of the polyhedrin promoter and p10 promoter, respectively in pFastBac Dual transfer vector (Thermo Fisher Scientific). HPIV3 L (M1–D2233) was fused with a C-terminal FLAG tag (DYKDDDDK) and 25 additional N-terminal residues MISNQQSDNGQKENIKNLGAKRARK, and P (M1–Q603) was fused with a C-terminal His$_6$ tag. Recombinant bacmid containing hPIV3 L–P genes was generated and isolated following the instruction manual of the Bac-to-Bac baculovirus expression system (Thermo Fisher Scientific). Viral stocks generated from purified bacmids were amplified and used for protein expression. 2.4 L of Sf21 cells (Thermo Fisher Scientific) were infected with amplified viruses to co-express hPIV3 L and P proteins at 27 °C for 48 h. The pellet of Sf21 cells expressing the hPIV3 L–P complex was resuspended through high-pressure homogenizer in the lysis buffer (50 mM Tris-HCl, pH 7.5, 500 mM NaCl, 10% glycerol, 6 mM MgSO$_4$, 1 mM dithiothreitol (DTT), 1% Triton X-100) supplemented with cOmplete™, EDTA-free protease inhibitor cocktail (Roche). After high-speed centrifugation at 50,000× g for 60 min at 4 °C, the supernatant containing the target proteins was loaded onto anti-FLAG G1 affinity resin (Genscript). The resin was washed using buffer A (20 mM Tris-HCl, pH 7.5, 300 mM NaCl, 5% glycerol, 6 mM MgSO$_4$, 1 mM DTT), and the bound proteins were eluted using 0.2 mg/ml FLAG peptide in buffer A. The eluted proteins were further purified using a size-exclusion chromatography (SEC) column (Superose 6 Increase 10/300 GL, GE healthcare) equilibrated with buffer B (20 mM Tris-HCl, pH 7.5, 300 mM NaCl, 1% glycerol, 6 mM MgSO$_4$, 1 mM DTT). The peak fractions were collected and analyzed by sodium dodecyl sulfate-polyacrylamide gel electrophoresis (SDS-PAGE). Fractions that contained the L–P complex were combined and concentrated to 1.2 mg/ml. The final sample was flash-frozen, and stored at −80 °C. The protein sample homogeneity was characterized by negative-stain EM. Molecular weight (MW) of the purified hPIV3 L–P complex was determined by analytical SEC using a Superdex 200 Increase 5/150 GL column (GE healthcare) with buffer B. The standard molecular markers (Cytiva, 28403842) were used for calibration. In addition, size-exclusion chromatography with multi-angle light scattering (SEC-MALS) was also used for MW determination of the complex. The purified hPIV3 L–P complex was loaded onto a XBridge™ Protein BEH SEC column (200 Å, 3.5 μm, 7.8 × 150 mm, Waters) using Breeze 2 high-performance liquid chromatography system (Waters) coupled with a DAWN MALS detector (Wyatt Technology) at 0.5 ml/min. The running buffer was the same as buffer B that was used in purification. Bovine serum albumin was used as a control.

### Fluorescence-based primer extension assay

For the determination of the hPIV3 L–P RdRp activity, a real-time primer elongation assay was established utilizing the fluorescence dye SYTO9 (Thermo Fisher Scientific), which binds to double-strand RNA but not single-strand RNA[27,47]. A 40-nt RNA oligonucleotide with the sequence of 5′-UUUUUUUUUUUUUUUUUUUUUAGUUCUUCUCUU-GUUUGGU-3′ was used as the template strand, and a 4-nt RNA oligonucleotide with the sequence of 5′-ACCA-3′ was used as the primer strand. To prepare the RNA duplex, 100 μM of both oligonucleotides were mixed at equal volume in the annealing buffer (10 mM Tris-HCl, pH 8.0, 25 mM NaCl and 2.5 mM EDTA) with 0.57 U/μl RiboLock RNase inhibitor (Thermo Fisher Scientific), denatured by heating to 94 °C for 5 min, and then slowly cooled to room temperature. The primer extension assay was performed in a 384-well plate (PerkinElmer). The reaction buffer contained 20 mM Tris-HCl, pH 8.0, 10 mM NaCl, 10 mM KCl, 6 mM MgCl$_2$, 0.01% TritonX-100, 1 mM DTT, 5% glycerol and 0.025 U/μl RiboLock RNase inhibitor. The purified hPIV3 L–P complex with a final concentration ranging from 0 to 100 nM was added into the reaction mixture containing 0.2 μM RNA duplex, 2 mM nucleoside triphosphates (ATP, GTP, CTP, and UTP at 0.5 mM each) and 0.125 μM SYTO9, to initiate the RNA elongation. The Infinite M1000 (Tecan) was employed to record the fluorescence signal using excitation wavelength and emission wavelength at 485 and 535 nm, respectively in real-time for 30 min at 25 °C. Data are representative of two independent experiments.

### Cryo-EM sample preparation and data collection

An aliquot of 3.5 μL of hPIV3 L–P complex at 0.8 mg/ml was applied to a freshly glow-discharged Quantifoil R1.2/1.3 300-mesh grid. The sample was immediately blotted at 4 °C and 100% relative humidity, then plunge-frozen in liquid ethane using Vitrobot Mark IV (Thermo Fisher Scientific), and finally stored in liquid nitrogen. Cryo-EM data were collected on a Titan Krios microscope (Thermo Fisher Scientific) under 300 kV, equipped with a K3 Summit direct electron detector (Gatan). Movie stacks were automatically recorded using AutoEMation[48] in the super-resolution mode at a nominal magnification of 81,000×, corresponding to a physical pixel size of 1.087 Å. The defocus was set from −1.5 to −2.0 μm. A total exposure dose of 50 e$^-$/Å$^2$ was fractionated into 32 frames for each movie stack. Finally, we obtained one cryo-EM dataset of hPIV3 L–P complex including a total number of 5800 movie stacks.

### Cryo-EM image processing

A flow chart of cryo-EM data processing is shown in Supplementary Fig. 1. All dose-fractionated movie stacks were motion-corrected with RELION's own motion-correction implementation[49], yielding micrographs of 1.087 Å pixel size. After contrast transfer function (CTF) estimation using cryoSPARC[50] Patch CTF, a total of 5693 micrographs were selected for subsequent processing. To generate templates for automatic particle picking, 642 micrographs were selected, and 589,198 particles were auto-picked using cryoSPARC's blob picker and extracted with a box size of 128 pixels after binning. After 2D classification, 100,387 particles were selected for 3D classification using Ab-Initio reconstruction in cryoSPARC, and three classes were generated as the initial reference models. Then 50 2D templates were projected from the model with clear structural features. For the dataset of hPIV3 L–P complex, 4511 micrographs were selected based on fitted resolution better than 4 Å, and a total of 3,288,374 particles were picked using templates generated previously and extracted with a box size of 150 pixels after binning. 636,487 particles were selected after two rounds of 2D classification based on the complex integrity. Then heterogeneous refinement was performed in cryoSPARC using previously generated Ab-Initio models. A subset of 373,487 particles from the class showing clear structural features was selected and re-extracted with a box size of 360 pixels without binning, and the resolution reached 3.3 Å after homogeneous refinement. For further classification, the full complex model and two erased models were used as 3D volume templates for heterogeneous refinement. Two main classes of

high quality (class 1 and 2, with and without a large extra blob of electron density, respectively) were subjected to the following refinements. After homogeneous refinement and local refinement performed in cryoSPARC, followed by CTF refinement, Bayesian polishing refinement, 3D classification and 3D auto-refine performed in RELION[49], a 2.7 Å cryo-EM density map was obtained for class 1 comprising 102,956 particles. Homogeneous refinement, local CTF refinement and non-uniform refinement performed using cryoSPARC yielded a 3.3 Å cryo-EM density map for class 2 comprising 137,294 particles. In order to further evaluate class 1 and see if we can observe more density of the second L–P copy, we attempted to use RELION's particle subtraction to subtract the signal of the "complete" L–P copy from the particles of class 1, re-center and re-extract the particles with a box size of 250 pixels for reconstruction. 3D classification without alignment (using the alignment information derived from previous processing) was performed in cryoSPARC to produce one class that contained CD′ and smeared density. Local refinement in cryoSPARC after erasing smeared density obtained a 4.2 Å map of CD′. The subtracted particles were also subjected to three rounds of 2D classification with new alignments in cryoSPARC, and several 2D classes comprising 54,669 particles were selected. Ab-Initio reconstruction and heterogeneous refinement were performed in cryoSPARC to produce two 3D classes. A subset of 31,643 particles from the class showing clear structural features was selected. Then a 3D initial model was generated in RELION, and the following 3D classification and 3D auto-refine were performed to obtain a 7.0 Å map of potential RdRp-PRNTase of L–P copy 2, while other regions that could not be aligned well were removed during reconstruction.

In addition, 708,311 particles of the 2D classes that had a bigger size or two potential ring-like L cores were selected from the excluded particles in the first round of 2D classification during class 1 and class 2 reconstruction. After three rounds of 2D classification using cryoSPARC, several 2D classes comprising 54,621 particles were observed with two featured ring-like L cores with potential bridging parts and appeared to be entire L–P dimers. However, we failed to reconstruct further for these classes.

## Model building and refinement

The PIV5 L–P structure (PDB entry: 6V85)[6] was used to guide the building of the atomic models of the hPIV3 L–P class 1 and 2. The starting model composed of RdRp-PRNTase domains of L, four copies of P-OD and single P-XD of PIV5 L–P complex was placed and rigid-body fitted well into the class 1 cryo-EM map using UCSF Chimera[51]. The CD and MTase-CTD domains of L were rigid-body docked separately with some rotation. Manual model building was carried out using Coot[52] and refinement of the coordinates was performed using phenix.real_space_refine[53]. One magnesium ion at the RdRp catalytic center could be built due to its well-resolved electron density and close distance (about 2 Å) to the side chain oxygen of L residue Asp773. For the large extra blob of electron density in class 1 map, the main chains were first traced based on the excellent continuity of the electron density and obvious secondary structure features, and bulky side chains visible in the density were utilized to determine the correct register of residues. Then the rough model was superimposed on the already built hPIV3 L–P structure, and the result showed that it shares a high structural similarity to the CD domain of hPIV3 L. Based on the critical hint, this region was unambiguously built into the cryo-EM map and assigned as the CD domain of a second L protein in class 1 model. The class 2 model was then built based on the class 1 model. The final hPIV3 L–P model of class 1 comprises: the full L protein residues except for the N-terminal 7 residues, Tyr611–Lys637, Leu1292–Met1299, Ile1693–Asp1706, Thr1745–Thr1762, Thr2095–Lys2113 and the C-terminal 23 residues; the P protein residues with four copies of OD domains (Asp435–Gly471, Ala434–Met467, Asp435–Gly472 and Asp435–Asp475 of subunits P1, P2, P3, and P4, respectively) and single

XD domain (Asn539–Gln603); and the CD domain of second L protein residues Asp1425–Leu1458 and Ile1470–Ile1687. The final hPIV3 L–P model of class 2 comprises nearly the same residues as class 1 except for the second CD domain. MolProbity[54] was used to validate the geometries of the final models and the statistics are given in Supplementary Table 1.

## Plasmids

Two minigenomes, hPIV3-RenLuc and hPIV3-CAT1/CAT2 were employed. The viral sequences within each were based on the hPIV3 JS strain (GenBank accession code: NC_001796) and they were designed based on a previously described hPIV3 minigenome[55]. The hPIV3-RenLuc minigenome contained in 3′ to 5′ order the hPIV3 55-nt leader region, 10-nt N gene start signal, 37-nt N 5′ non-translated region, a vaccinia virus terminator sequence (AAAAAUA in negative sense), 2 additional nucleotides and an *Xba*I restriction site, the *Renilla* luciferase open reading frame (ORF), followed by a *Kpn*I restriction site, 62 nt of the L 3′ non-translated region, 11-nt L gene end signal and 44-nt trailer region. The hPIV3-CAT1/CAT2 dicistronic minigenome contained the same elements as the hPIV3-RenLuc minigenome except that the *Renilla* luciferase ORF is replaced by the first 520 nt of the chloramphenicol acetyltransferase (CAT) ORF (CAT1), the 30-nt hPIV3 N-P gene junction, and the last 142 nt of the CAT ORF (CAT2) in negative sense. Both minigenomes followed the rule of six[56]. Each hPIV3 minigenome cassette was flanked with a T7 promoter at the 5′ end, with two G nucleotides between the promoter and the trailer region, and a hepatitis delta virus ribozyme sequence to generate the RNA template 3′ end. To generate the support plasmids, the L and P ORFs were derived from the same pFastBac Dual recombinant plasmid that was used for protein expression, described above. The 25 amino acid N-terminal addition was removed from L, and the L ORF was inserted as the FLAG-tagged or untagged form. The His$_6$ tag was removed from P. A codon-optimized version of the open reading frame of hPIV3 N gene (Gene ID: 911955) was synthesized by Synbio. The open reading frames of L, P and N were each cloned into a pTM1 vector[57] that had been modified to contain a 17-nt poly A sequence (and flanking random sequence) between the *Xho*I and *Sal*I restriction sites. Each ORF was inserted into the *Nco*I and *Spe*I sites such that they immediately follow the internal ribosome entry site and are followed by the poly A and T7 terminator sequences. To perform mutagenesis of L, two L fragments spanning the L ORF were subcloned into a shuttle vector. Mutagenesis of the L gene was performed using the Q5 Site Directed Mutagenesis kit (NEB) and all sequences that were subject to PCR amplification were sequence verified. The sequenced fragment was reinserted into the pTM1 vector, and the integrity of the ligation sites were confirmed by sequence analysis. As an internal control for luciferase assays, the Firefly luciferase ORF was cloned into the poly A modified pTM1 vector, to generate pTM1-FF. Sequences of the plasmids are available upon request.

## Minigenome assays

Minigenome transcription and replication were reconstituted in human epithelial type 2 (HEp-2) cells as described previously[55]. Briefly, HEp-2 cells (ATCC) were cultured in Opti-MEM I reduced serum medium (Gibco) supplemented with 2% fetal bovine serum (Gibco) and GlutaMax (Gibco). Monolayers of HEp-2 cells grown in 6-well plates to 75-90% confluence were transfected with 400 ng/well of pTM1-hPIV3 N, 400 ng/well of pTM1-hPIV3 P, 50 ng/well of pTM1-hPIV3 L, and 400 ng/well of minigenome hPIV3-RenLuc or hPIV3-CAT1/CAT2, as well as 40 ng/well of pTM1-FF in the case of transfections with hPIV3-RenLuc. Transfections were performed using Lipofectamine 3000 (Thermo Fisher Scientific) in Opti-MEM I according to the manufacturer's instructions. 15 min following transfection, cells were infected with modified vaccinia virus Ankara-T7[58] using the amount of virus that had been empirically determined to give optimum minigenome

expression. Cells were incubated at 37 °C in 5% $CO_2$ for 16 h, after which the transfection mix was replaced with 1 ml/well of Opti-MEM I containing 2% FBS. Cells were harvested ~40–48 h post-transfection.

## Luciferase assays

Cells from each well of a 6-well plate were harvested into 250 μL of passive lysis buffer from the dual luciferase reporter assay system (Promega). 10 μL of 10-fold diluted lysate was mixed with 50 μL of Luciferase Assay Reagent II. Firefly and *Renilla* luciferase activities were measured by adding 50 μL of Stop & Glo reagent on an Omega Luminometer (BMG Labtech). *Renilla* luciferase signals were normalized to Firefly luciferase signals to account for transfection efficiency, expressed as relative luciferase units (RLU), and the resulting values were then normalized to the corresponding signal of the FLAG-tagged WT L. For the luciferase-based complementation assay, the signal of the no L sample, as the background was subtracted from each sample prior normalization.

## Western blot

20 μL of cell extract obtained with the passive lysis buffer were analyzed by SDS-PAGE (8% polyacrylamide) and transferred to nitrocellulose membranes (Amersham Protran premium 0.45 μm NC). After transfer, membranes were blocked with 5% milk in PBS-T (phosphate-buffered saline with 0.1% Tween 20) with constant agitation. Membranes were incubated in the same buffer with a rabbit anti-FLAG monoclonal antibody (Cell Signaling Technology, 14793 S, clone D6W5B) diluted 1:1,000 to detect FLAG-tagged L proteins, and mouse anti-GAPDH monoclonal antibody (Proteintech, 60004-1-Ig, clone 1E6D9) diluted 1:2,000 to detect GAPDH as a loading control. After washing with several changes of PBS-T, membranes were incubated with donkey anti-rabbit IRDye 680RD (LI-COR, 926-68073) and goat anti-mouse IRDye 800CW antibodies (LI-COR, 926-32210) diluted 1:20,000 in PBS-T for 1 h at room temperature. Signals were detected and quantified using the Odyssey DLx imaging system (LI-COR).

## RNA analysis

Total intracellular RNA was isolated using an RNeasy kit (Qiagen) according to the manufacturer's instructions. Northern blot transfer, probe preparation, and probe hybridization were performed as described previously[59]. Negative and positive sense $^{32}P$-labeled CAT-specific riboprobes were synthesized with T7 RNA polymerase as described previously[60]. The CAT1 mRNA and antigenome RNA on Northern blots were identified by migrating RNA alongside a molecular weight ladder (Dynamarker prestain marker for RNA high). Following hybridization, the membrane was exposed to autoradiographic film (GE Healthcare), the film was aligned to the blot with the colored markers, and the positions of the markers were marked onto the film, allowing us to confirm the sizes of the RNAs. Subsequent Northern blots were analyzed by phosphorimager analysis: data acquisition was made on phosphorimager (Personal Molecular Imager, Bio-Rad) using the associated software Quantity one 1-D (Bio-Rad). The background signal was subtracted from each sample value, and the resulting value was then normalized to the corresponding RNA signal of the FLAG-tagged WT L.

## Figure preparation

UCSF Chimera[51] and Pymol (https://pymol.org/2/) were used for structure visualization and figure generation. Multiple sequence alignments were performed with PROMALS3D[61] (http://prodata.swmed.edu/promals3d/promals3d.php) combined with secondary structures and tertiary structures, and the alignment results were displayed with ESPript[62]. The buried interface was calculated by PDBePISA[63].

## Reporting summary

Further information on research design is available in the Nature Portfolio Reporting Summary linked to this article.

## Data availability

All data supporting the findings of this study are available in the manuscript and the supplementary materials. The cryo-EM density maps generated in this study have been deposited in the Electron Microscopy Data Bank (EMDB) under accession codes EMD-37130 (hPIV3 L–P class 1, incomplete dimeric form) and EMD-37131 (hPIV3 L–P class 2, monomeric form). The corresponding atomic coordinates have been deposited in the Protein Data Bank (PDB) under accession codes 8KDB (hPIV3 L–P class 1, incomplete dimeric form) and 8KDC (hPIV3 L–P class 2, monomeric form). The previously published structures used for comparison and analysis in this study are available in the PDB under the following accession codes: 6PZK, 6QCX, 6U1X, 6U5O, 6UEB, 6V85, 7BV2, 7YES, 7YOU, 7YOV, 8JSN and 8SNX. Source data are provided with this paper.

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

## Acknowledgements

The authors appreciate the cryo-EM facility at Westlake University for the assistance in cryo-EM data acquisition. We would like to thank Zhanyu Ding from Shanghai YueXin Life-science Information Technology Co. Ltd. for suggestions and help in cryo-EM data processing. We also thank the support of Minqi Gao and Xiaoyi Ji from Wuxi Biortus Biosciences Co. Ltd. in biochemical assay. The project was funded by a Roche Post-doctoral Fellowship with the project number RPF573.

## Author contributions

L.W., L.G., X.H., R.F. and S.C. conceived the study. J.X., M.W. and X.Y. carried out protein expression and purification, electron microscopy data collection and procession, and structure building and refinement, supervised by S.C. M.O. established the minigenome system and designed and performed functional assays, with guidance and training from B.L. and R.F. G.Z., D.W., and Z.L. performed biochemical assays. J.X., M.O., L.W., G.Z., D.W., B.L., T.N., R.F. and S.C. analyzed and interpreted data. J.X., M.O., L.W., X.H., R.F. and S.C. wrote the manuscript with input from all authors.

## Competing interests

J.X., L.W., G.Z., D.W., Z.L., T.N., L.G., X.H., and S.C. are either current or former employees of Roche Innovation Center Shanghai or Basel. The project was funded by Roche in a collaboration with R.F. at the Boston University Chobanian & Avedisian School of Medicine. R.F. also has a sponsored research agreement with Merck & Co., Inc. M.O. and B.L. are either current or former members in R.F.'s lab. The remaining authors declare no competing interests.
