## [Peer Review File · Nature Communications]

Structural basis for dimerization of a paramyxovirus polymerase complexREVIEWER COMMENTS

Reviewer #1 (Remarks to the Author):

Xie and colleagues submit a manuscript that describes the novel cryoEM derived structure of the L and P proteins from human parainfluenza virus type 3 (hPIV3). Over the last eight years, structures of L proteins from non-segmented negative strand RNA viruses, nsNSRV, (and some segmented viruses) have been more forthcoming due to advances in sample preparation and amenability to cryoEM. The drive of structures has provided insight to conformational variability of L, diversity of P-binding sites, and snapshots of template-binding. In some previous studies, multimeric complexes of L and P have been noted, but to date, there has been no high-resolution detail of these assemblies, and moreover there is no understanding of their relevance. This manuscript provides a detailed analysis of the L and P structure, advancing the field by noting the dimeric L-L interface. Through comparison to the polymerases of other recently determined nsNSRV L proteins and Flu Pol, and the positive strand SARS-CoV-2 Pol, the high-resolution structure is able to provide new potential mechanistic information. Overall, the manuscript is an excellent study of polymerase structure that is supported by sound functional studies, including mutational analysis and in cellular minigenome assays. This reviewer finds no shortcomings with the written portion of the manuscript.

Two minor notes on presentation are provided:

Fig 2. a/b: it's difficult to distinguish the GDN residues in a background carbon color that is the same as the cartoon. In the current state, presentation of the residues is of diminished value.

Fig 2. c/d: the residue labels especially with white shadowing crowd the field of view, sometimes obscuring the residues of note.

Reviewer #2 (Remarks to the Author):

In this study the authors have solved cryo-EM structures of the L-P complex from hPIV3 which are nicely validated by a set of functional experiments. The major finding of the work is one structure that includes the connector domain (CD) of a second copy of L-P, thereby implying a dimerization interface for the polymerase complexes. Dimerization of viral polymerases has been much studied in influenza viruses and is an important mechanistic piece in the puzzle of L function. The structural data appear to be of high quality and close inspection of the PDB Validation Reports shows no obvious issues. The subject matter of the manuscript is important and the experiments have been well-executed. It is unfortunate that only a single domain of the second polymerase could be resolved. As this is an important research question, I hope that the authors consider attempting some additional processing approaches (see comment 2 below) to potentially resolve more of the second L-P, thereby further elevating the work. However, the existing work is already sound and impactful enough for Nature Communications and acceptance of the manuscript should not hinge on this request.

My specific comments below.

1) The presentation as a structure of “of a dimeric paramyxovirus polymerase complex” may be seen as slightly misleading. While the sample is clearly dimeric, the structure doesn’t resolve a dimer but rather a monomer bound the CD of its partner. I would therefore appreciate a slightly toned-down or qualifying title. Perhaps including “structural basis of dimerization”?

2) While it is clear that two L-P copies are present, the authors observe only the CD of the second copy. Could the authors consider attempting reconstruction of the second copy by recentering the origin of their particle set of 102,956 dimeric particles on the CD', re-extracting particles (perhaps in a smaller boxsize), and then attempting ab-initio and heterogeneous refinement jobs with these new particles. In their current processing approach, the particle alignments seem to be driven by one copy of L-P, to the detriment of resolving the second. With some clever processing approaches, reducing the influence of L-P copy 1 on the alignment, I would expect at least a bit more of the second L-P to be resolvable than it is currently. A composite map of the first and second L-P copies could then be produced. However, I do understand that this request might entail a substantial amount of additional work and will thus not insist on it if the authors feel it is beyond their capacity. The work as it stands is of excellent quality regardless.

3) In the first paragraph of the results, could the authors please state more clearly that only the CD of a second L-P copy is resolved, while the rest of this L-P is disordered, presumably due to structural heterogeneity?

4) Lines 106-133: The authors describe in detail the domain, intrusion loop, and priming loop differences between hPIV3, PIV5, and VSV. What I am missing here a bit is a conclusion. How do these distinct conformations fit into the idea introduced at the end of the manuscript that this might be an encapsidating polymerase?

5) The biophysical data presented in Fig. 5 indicates that the majority of the author’s sample is dimeric L-P. I assume this is not generally the case in preparations of nsNSV L-P complexes? Do the author’s think this is specific for hPIV3? Could they speculate why this might be the case here and less so for other L-Ps from nsNSVs?

6) As the interaction with the CD from a second L-P copy is the major finding of the paper, it would be nice to show a partial multiple sequence alignment including the most important involved residues for this, in the main text.

7) The functional model for a nsNSV L-P dimer is important and should be moved from the supplement to the main text.

8) Fig. 2c-f: The panels c-f are unfortunately busy and complex. With so many residues, colors, labels, and features displayed simultaneously it is difficult to see what is going on. Could the authors simplify the panels and make them easier to visually understand? Perhaps side-by-side structures might be preferable to aligned structures? Other suggestions could be the use of thicker stick representations for side chains, and not using too many different colors at once.

Minor comments:

1) line 67: “Interestingly, a large extra blob of electron density, not reported in previous L-P structures, was unambiguously observed near the intact L protein and was successfully assigned as the CD domain of a second L protein”

It would be nice if the authors could highlight the second CD in Fig. 1 more clearly, for instance with an arrow.

2) Line 101: “Coupled with this EBOV L-RNA structure, our results facilitate the understanding of other aspects of nsNSV RdRp active site, including the catalytic Mg²⁺, the +1 site and interactions with template-product RNA duplex.”

This is a very general sentence. The authors could either remove it or make it more specific.

3) Fig.7: It would be great if the authors could add a panel to this figure highlighting the locations of the mutated residues in the structure, such that the reader can visually assess why these were chosen.

Reviewer #3 (Remarks to the Author):

Xie et al determined the structure of human parainfluenza 3 polymerase (hPIV3 L) bound to its phosphoprotein co-factor (hPIV3 P) by cryo-EM. hPIV3 is a non-segmented negative-stranded RNA virus (nsNSV), a viral group that contains several serious pathogens. Its polymerase is an essential enzyme involved in viral genome replication and transcription. Determining structures of this enzyme is therefore very important to understand the essential structural elements that are important for replication and transcription. It also opens new avenues for future rational design of

drugs. The hPIV3 polymerase structure adds to the structural information obtained during the last few years on nsNSV polymerases. Although the conformation visualized is similar to the one recently observed on Newcastle disease virus polymerase-phosphoprotein complex (NDV L-P), interesting elements are visualized such as a beta-strand latch that stabilizes the C-terminal domain of the polymerase. Xie et al also visualize the priming loop and HR protrusion loop in new positions and the results therefore significantly add to the current knowledge on nsNSV polymerases. Both L-P binding and beta-strand latch/CTD interactions are well described, and their importance is carefully analysed by mutations in reverse-genetic systems.

If this study is solid in many aspects, it suffers from a major issue that I will require to be modified: the authors claim having determined the cryo-EM structure of a dimer, as written in the title, in the abstract and in the results. However they only visualize one polymerase bound to the connector domain (CD) of another polymerase. Only 12% of the second polymerase is seen. Claiming visualizing a dimer is misleading and absolutely needs to be modified to fit with reality.

I therefore request major modifications:

1: the authors should rephrase the title, the abstract, the last paragraph of the introduction and the result to clearly state that they visualize one entire polymerase + a CD domain.

2: It would be less misleading if the biophysical characterization of the sample (analytical SEC and MALLS) and the cryo-EM entire flowchart (see point 3) was shown at the beginning of the result section. The differences between the SEC-MALLS and the 2D class averages is very intriguing. Were cryo-EM and SEC-MALLS done in the same condition? During cryo-EM freezing, one cannot exclude that the buffer significantly changes during the blotting time. This might cause degradation or disrupt the dimers. Could the author try to perform MALLS in a different buffer (higher NaCl and low NaCl concentration) to see if the dimer is stable and if degradation could have occurred? If no degradation occurs and if hPIV3 L remains dimeric, then the visualization of one monomer+CD' is likely to be due to flexibility as stated by the authors.

3: The authors show class averages of monomer+CD' and class averages of dimers in two separate figures. It would be nice to have the flowchart of the processing that includes both the monomers+CD (SF1) and the supposed dimers (Fig.5c). Some of the 2D class averages currently shown in Fig. 5c are very likely to correspond to particles that were close on the micrographs but that are monomeric. This is obvious when comparing with 2D class averages of real polymerase dimers from other NSV. If the authors want to put this figure in Supplementary information, they should say how many particles are in these classes and clearly mention which ones may not be real dimer.

4: lines 245-248: the model of Supplementary Figure 10 with as asymmetric dimer and its structural analysis should be removed. Indeed, the CD-MTase-CTD (at least) are able to undergo drastic conformational changes and one cannot infer their potential position without experimental data. For nsNSV, the structures during replication and transcription are still missing and therefore nobody knows the conformation during activity. More advanced analysis on sNSV polymerases (Influenza, Lassa, SFTSV, La Crosse polymerases) show drastic conformational changes during replication and transcription. It is therefore very likely that large conformational rearrangement also occur for nsNSV L. The authors here go too far, their model does not have the necessarily basis and is therefore very likely to be incorrect. It would thus ask to remove this model and I would strongly recommend to state why the structure of an entire dimer cannot be inferred.

5: the discussion on the replicase-encapsidase model for nsNSV (lines 249-262) is speculative. As said on point 4, Supplementary Figure 10 is not a structure, it is a speculative model that is not supported by experiment. In Supplementary Figure 11, the authors go further and propose that the replicase-encapsidase of nsNSV is similar to the one of Influenza. However, it is not known if the mechanism of encapsidation is similar for nsNSV. Indeed in Influenza, the 5' end of the nascent RNA that is newly synthesized by the replicase binds specifically in a conserved binding pocket of the encapsidase. This pocket has not been identified for nsNSV. This is acknowledged by the authors who put a question mark in their Supplementary Fig. 11 for the 5' of the nascent RNA. This is a central point of the replicase-encapsidase mechanism in Influenza. Therefore one cannot infer a model for non-segmented NSV L without structural data of an entire dimer. I would therefore recommend to be very careful in this discussion. I would recommend basing it on real data, i.e. analysing the charges on L+CD' as in Fig. 7e.

6:line 264: the data does not reveal the entire L-L dimer interface. It reveals the interface between L and the CD' domain. This needs to be corrected.

In a nutshell, the data that is based on experiment is carefully analysed and meaningful. But the structure is not hPIV3 L dimer, it is hPIV3 L monomer+CD'. Discussion about the function of a dimer that is not structurally determined is therefore highly speculative.

Minor point:

1: line 62: two bands are seen for P in Fig 1b. It would be interesting to know what are these bands. Would it be possible to analyse them by mass spectrometry to see if it is a truncation or a phosphorylation?

2: lines 66-67: it would be interesting to know the percentage of identity between hPIV3 L, PIV5 L and NDV L. This is particularly important also because for NDV L as the conformation of hPIV3 L-P is similar to the one of NDV L-P.

3: lines 101-103: it would be interesting to overlay (or show on two neighboring panels) hPIV3 RdRp active site and ebola RdRp bound to single-stranded template RNA to illustrate the sentence written.

4: lines 139-140: it would be interesting to add a figure that shows why the short beta-sheet formed with the beta-strand latch should be susceptible to disruption at elongation.

5: lines 145-146: stabilization of the C-terminal region/core interaction through latch/lariat/beta-hairpin strut is also observed for segmented NSV polymerases (La Crosse, SFTSV and Lassa notably). This could be added as it seems to be important for stabilization of the CTER of very different NSV L.

6: lines 152-156. This is difficult to follow because the residues mentioned in the text are not always visible on the related figure. This should be corrected.

REVIEWER COMMENTS

Reviewer #1 (Remarks to the Author):

Xie and colleagues submit a manuscript that describes the novel cryoEM derived structure of the L and P proteins from human parainfluenza virus type 3 (hPIV3). Over the last eight years, structures of L proteins from non-segmented negative strand RNA viruses, nsNSRV, (and some segmented viruses) have been more forthcoming due to advances in sample preparation and amenability to cryoEM. The drove of structures has provided insight to conformational variability of L, diversity of P-binding sites, and snapshots of template-binding. In some previous studies, multimeric complexes of L and P have been noted, but to date, there has been no high-resolution detail of these assemblies, and moreso there is no understanding of their relevance. This manuscript provides a detailed analysis of the L and P structure, advancing the field by noting the dimeric L-L interface. Through comparison to the polymerases of other recently determined nsNSRV L proteins and Flu Pol, and the positive strand SARS-CoV-2 Pol, the high-resolution structure is able to provide new potential mechanistic information. Overall, the manuscript is an excellent study of polymerase structure that is supported by sound functional studies, including mutational analysis and in cellular minigenome assays. This reviewer finds no shortcomings with the written portion of the manuscript.

Two minor notes on presentation are provided:

Fig 2. a/b: it's difficult to distinguish the GDN residues in a background carbon color that is the same as the cartoon. In the current state, presentation of the residues is of diminished value.

Fig 2. c/d: the residue labels especially with white shadowing crowd the field of view, sometimes obscuring the residues of note.

Response: Thanks for your recognition of our study and for your instructive suggestions! In the revised manuscript, we added an arrow to indicate the GDN residues and moved the label not too close to the cartoon in Fig. 2a and b so that they can be distinguished easily. For Fig 2. c/d, we enlarged the structure pictures, colored hPIV3 RdRp only in cyan instead of different colors, used transparent background, and adjusted the label positions of the residues to make the figures more clear.

Reviewer #2 (Remarks to the Author):

In this study the authors have solved cryo-EM structures of the L-P complex from hPIV3 which are nicely validated by a set of functional experiments. The major finding of the work is one structure that includes the connector domain (CD) of a second copy of L-P, thereby implying a dimerization interface for the polymerase complexes. Dimerization of viral polymerases has been much studied in influenza viruses and is an important mechanistic piece in the puzzle of L function. The structural data appear to be of high quality and close inspection of the PDB Validation Reports shows no obvious issues. The subject matter of the manuscript is important and the experiments have been well-executed. It is unfortunate that only a single domain of the second polymerase could be resolved. As this is an important research question, I hope that the authors consider attempting some additional processing approaches (see comment 2 below) to potentially resolve more of the second L-P, thereby further elevating the work. However, the existing work is already sound and impactful enough for Nature Communications and acceptance of the manuscript should not hinge on this request.

My specific comments below.

1) The presentation as a structure of “of a dimeric paramyxovirus polymerase complex” may be seen as slightly misleading. While the sample is clearly dimeric, the structure doesn’t resolve a dimer but rather a monomer bound the CD of its partner. I would therefore appreciate a slightly toned-down or qualifying title. Perhaps including “structural basis of dimerization”?

Response: Thanks for your instructive suggestion. The title has been changed to “Structural basis for dimerization of a paramyxovirus polymerase complex” as you suggested.

2) While it is clear that two L-P copies are present, the authors observe only the CD of the second copy. Could the authors consider attempting reconstruction of the second copy by recentering the origin of their particle set of 102,956 dimeric particles on the CD', re-extracting particles (perhaps in a smaller box size), and then attempting ab-initio and heterogeneous refinement jobs with these new particles. In their current processing approach, the particle alignments seem to be driven by one copy of L-P, to the detriment of resolving the second. With some clever processing approaches, reducing the influence of L-P copy 1 on the alignment, I would expect at least a bit more of the second L-P to be resolvable than it is currently. A composite map of the first and second L-P copies could then be produced. However, I do understand that this request might entail a substantial amount of additional work and will thus not insist on it if the authors feel it is beyond their capacity. The work as it stands is of excellent quality regardless.

Response: Thanks very much for your instructive suggestion. Following the advice provided, we reconstructed L-P copy 2 in class 1 by subtraction of the “complete” L-P copy 1. Some interesting results were obtained and are summarized in a new supplementary figure 2. The analysis has been added into the first section of Results and Discussion in the revised manuscript (lines 81–98). After subtraction of L-P copy 1, 3D classification with good CD' alignments inherited from previous refinement of class 1 yielded a large smeared density except for CD', indicating the potential missing parts of the second copy of L-P (Fig. S2a/b). For the CD' part, focused refinement after erasing the smeared density obtained a 4.5 Å map. It is nearly the same as the original CD' map of class 1 except for the visible density of the loop (Lys1459–Asp1469) (Fig. S2c), reflecting the flexibility flanking the CD' with long loops in the second copy of L-P. 2D classification obtained classes with apparent ring-like RdRp-PRNTase core of L instead of featured CD', which may be caused by the new alignments with the larger RdRp-PRNTase core compared to that of CD'. After Ab-initio, heterogeneous refinement, and following refinements, we obtained a map of potential RdRp-PRNTase of the second L-P at a low resolution of about 7.0 Å (Fig. S2d). This map could overall be overlaid with the RdRp-PRNTase map and structure of class 1 and has the typical holes corresponding to the proposed template entry, template exit and RNA tunnel. Thus, we could reconstruct RdRp-PRNTase and CD domains of the second L in the hPIV3 L-P complex dimer separately, but their relative orientation could not be determined due to structural heterogeneity of the second copy of L-P.

3) In the first paragraph of the results, could the authors please state more clearly that only the CD of a second L-P copy is resolved, while the rest of this L-P is disordered, presumably due to structural heterogeneity?

Response: Thanks for your suggestion! We have corrected the text (lines 75–76, 91 and 96–98 in the revised manuscript).

4) Lines 106-133: The authors describe in detail the domain, intrusion loop, and priming loop differences between hPIV3, PIV5, and VSV. What I am missing here a bit is a conclusion. How do these distinct conformations fit into the idea introduced at the end of the manuscript that this might be an encapsidating polymerase?

Response: Thanks for your comment! We would like to express two conclusions that hPIV3 L adopts a distinct conformation from other reported nsNSV structures; and the conformations of priming and intrusion loops appear coupled with the rearrangements of the flexible C-terminal appendages of L. As it was not stated clearly in our previous version, we modified this part and added the second conclusion clearly at the beginning of this paragraph (lines 133–137 in the revised manuscript). In addition, based on this, we proposed that in the polymerase' priming loop and intrusion loop are retracted and pushes away the CD during RNA elongation with dramatic movement of the adjacent C-terminal MTase-CTD, resulting in a dynamic structure (in the figure legend of Fig. S14). However, we cannot link this information to encapsidating polymerase here. We have added text into the document to convey this point: "Although it seems likely that the different structures described represent different conformational states, how these different conformations correlate to different stages of transcription versus replication remains unknown." (lines 159–162).

5) The biophysical data presented in Fig. 5 indicates that the majority of the author's sample is dimeric L-P. I assume this is not generally the case in preparations of nsNSV L-P complexes? Do the authors think this is specific for hPIV3? Could they speculate why this might be the case here and less so for other L-Ps from nsNSVs?

Response: We are not sure whether the prepared L-P polymerase complexes in published papers did not include dimers as the gel filtration profile and the molecular weight information of the purified polymerase complexes were not usually provided. From the current publications, we know that VSV and RABV had 2D classes of dimers observed under negative-stain EM, but only monomeric L-P complex structures of RABV and VSV were reconstructed by cryo-EM. RSV and NDV papers provided SEC profiles, but without MW markers, however NDV may be a dimer based on the SEC profile.

Multiple sequence alignment showed the residues on L-L interface are conserved to some extent or some comparable interactions appear to exist. Therefore, we cannot exclude the conservation of L-L dimerization across paramyxoviruses and even other nsNSVs. (lines 255–263)

It seems that particle or structural heterogeneity is the main cause for the reconstruction failure of entire polymerase dimers by cryo-EM. The heterogeneity level may be different between different polymerases, especially when lacking optimal length and sequence of RNA, which may stabilize the dimers. It is to be noted that the expression and purification of the hPIV3 L-P complex samples is not always reproducible and the protein complexes were prone to aggregation.

6) As the interaction with the CD from a second L-P copy is the major finding of the paper, it would be nice to show a partial multiple sequence alignment including the most important involved residues for this, in the main text.

Response: Thanks for your suggestion! The interaction with CD from a second L-P copy is really the major finding of the paper, and the multiple sequence alignment of the residues on the interface should be important. Considering the involved residues are widely distributed on L, and it is not easy to use a partial

multiple sequence alignment in the main figure to summarize the information, we still would like to put this sequence alignment in the supplementary Fig. 4 with the full-length L sequence alignment. However, we have add detailed conservation analysis of the interface and conclusion in the main text of the revised manuscript to highlight the importance of the interface (lines 255–263).

7) The functional model for a nsNSV L-P dimer is important and should be moved from the supplement to the main text.

Response: Thanks for your suggestion! We agree with you that the functional model for a nsNSV L-P dimer is important. However, our study is the starting of the atomic structural and functional analysis of nsNSV L-P polymerase dimers and so the model that is presented is somewhat speculative at this stage as noted by reviewer 3. For this reason, we think it is more appropriate to retain the functional model as a supplement figure, with a clear statement that it is a speculative model.

8) Fig. 2c-f: The panels c-f are unfortunately busy and complex. With so many residues, colors, labels, and features displayed simultaneously it is difficult to see what is going on. Could the authors simplify the panels and make them easier to visually understand? Perhaps side-by-side structures might be preferable to aligned structures? Other suggestions could be the use of thicker stick representations for side chains, and not using too many different colors at once.

Response: Sorry for the complicated representation. In the revised manuscript we enlarged the structure pictures, used fewer colors and elements, and used thicker stick representations and transparent background to improve the clarity of the figure.

Minor comments:

1) line 67: “Interestingly, a large extra blob of electron density, not reported in previous L-P structures, was unambiguously observed near the intact L protein and was successfully assigned as the CD domain of a second L protein” It would be nice if the authors could highlight the second CD in Fig. 1 more clearly, for instance with an arrow.

Response: Thanks for your suggestion! A dotted circle has been added in Fig. 1d and e to highlight the second CD.

2) Line 101: “Coupled with this EBOV L-RNA structure, our results facilitate the understanding of other aspects of nsNSV RdRp active site, including the catalytic Mg²⁺, the +1 site and interactions with template-product RNA duplex.”

This is a very general sentence. The authors could either remove it or make it more specific.

Response: Thanks for your suggestion! We modified this text (lines 123–128) and added a supplementary Fig. 5 to make it more specific in the revised manuscript.

3) Fig.7: It would be great if the authors could add a panel to this figure highlighting the locations of the mutated residues in the structure, such that the reader can visually assess why these were chosen.

Response: Thanks for your suggestion! It is a good idea to show the locations of the mutated residues and state why they were chosen. We have added an explanation of the mutations. As the mutated residues are from the L-L dimer interface and clearly depicted in Fig. 6, we did not prepare an additional panel for Fig. 7 but instead referred the reader to Fig. 6.

Reviewer #3 (Remarks to the Author):

Xie et al determined the structure of human parainfluenza 3 polymerase (hPIV3 L) bound to its phosphoprotein co-factor (hPIV3 P) by cryo-EM. hPIV3 is a non-segmented negative-stranded RNA virus (nsNSV), a viral group that contains several serious pathogens. Its polymerase is an essential enzyme involved in viral genome replication and transcription. Determining structures of this enzyme is therefore very important to understand the essential structural elements that are important for replication and transcription. It also opens new avenues for future rational design of drugs. The hPIV3 polymerase structure adds to the structural information obtained during the last few years on nsNSV polymerases. Although the conformation visualized is similar to the one recently observed on Newcastle disease virus polymerase-phosphoprotein complex (NDV L-P), interesting elements are visualized such as a beta-strand latch that stabilizes the C-terminal domain of the polymerase. Xie et al also visualize the priming loop and HR protrusion loop in new positions and the results therefore significantly add to the current knowledge on nsNSV polymerases. Both L-P binding and beta-strand latch/CTD interactions are well described, and their importance is carefully analysed by mutations in reverse-genetic systems.

If this study is solid in many aspects, it suffers from a major issue that I will require to be modified: the authors claim having determined the cryo-EM structure of a dimer, as written in the title, in the abstract and in the results. However they only visualize one polymerase bound to the connector domain (CD) of another polymerase. Only 12% of the second polymerase is seen. Claiming visualizing a dimer is misleading and absolutely needs to be modified to fit with reality.

I therefore request major modifications:

1: the authors should rephrase the title, the abstract, the last paragraph of the introduction and the result to clearly state that they visualize one entire polymerase + a CD domain.

Response: Thanks for your comment! We have changed the title as you and reviewer #2 suggested, and modified other related text in the revised manuscript (lines 20–23, 54–56, 75–76 and 96–98).

2: It would be less misleading if the biophysical characterization of the sample (analytical SEC and MALLS) and the cryo-EM entire flowchart (see point 3) was shown at the beginning of the result section. The differences between the SEC-MALLS and the 2D class averages is very intriguing. Were cryo-EM and SEC-MALLS done in the same condition? During cryo-EM freezing, one cannot exclude that the buffer significantly changes during the blotting time. This might cause degradation or disrupt the dimers. Could the author try to perform MALLS in a different buffer (higher NaCl and low NaCl concentration) to see if the dimer is stable and if degradation could have occurred? If no degradation occurs and if hPIV3

L remains dimeric, then the visualization of one monomer+CD' is likely to be due to flexibility as stated by the authors.

Response: Thanks for your comments. The cryo-EM and SEC-MALLS assays were done in the same buffer, buffer B that were used in purification (line 358). We have reconstructed L-P copy 2 in class 1 by subtraction of the “complete” L-P copy 1 as reviewer #2 suggested, and obtained a map of potential RdRp-PRNTase of the second L-P at a low resolution of about 7.0 Å. The results showed that the structure that we solved is part of a complete L-P dimer, and that the reason for the failure of 3D reconstruction to get a complete dimeric structure is most likely due to structural heterogeneity of the second L-P copy. We have added these data in the first section of Results and Discussion.

Also, thank you for your suggestions of moving the results of biophysical characterization of the protein sample (analytical SEC and MALLS) to the beginning of the result section in the manuscript. We thought the current manuscript structure would help authors better understand the overall research plan we had conducted. At the beginning, we performed cryo-EM data processing that obtained two classes of high-resolution map and we didn't realize one of them is a polymerase dimer until we solved the class 1 structure of one L-P copy bound with CD' of the second copy. Afterwards, more experiments including biophysical characterization and cryo-EM data re-processing aimed to resolve potential entire dimers (suggested by reviewer 2, supplementary figure 2) were performed followed by functional studies to validate the dimer structure. For this reason, we think it is appropriate to retain the current manuscript flow, and put the background, results and discussions about the L-P polymerase dimerization mainly in the section “Structural definition of the L-L interaction”.

3: The authors show class averages of monomer+CD' and class averages of dimers in two separate figures. It would be nice to have the flowchart of the processing that includes both the monomers+CD (SF1) and the supposed dimers (Fig.5c). Some of the 2D class averages currently shown in Fig. 5c are very likely to correspond to particles that were close on the micrographs but that are monomeric. This is obvious when comparing with 2D class averages of real polymerase dimers from other NSV. If the authors want to put this figure in Supplementary information, they should say how many particles are in these classes and clearly mention which ones may not be real dimer.

Response: Thanks for your suggestion! We re-selected the 2D class averages of the potential polymerase dimers and discarded the misleading classes, and indicated the proposed bridging parts between two L cores of each class by light blue arrows in the revised Fig. 5c. We added more details of the processing of this part including the particle number in Methods.

Considering that the SEC-MALS result and the 2D class averages of the supposed dimers are the important data for dimerization of hPIV3 L-P complex which is the major finding of our paper, we put them together in separate Fig. 5, while Fig. S1 mainly depicts the flowchart of the processing and the validation of the two main classes of structures with high resolution.

4: lines 245-248: the model of Supplementary Figure 10 with as asymmetric dimer and its structural analysis should be removed. Indeed, the CD-MTase-CTD (at least) are able to undergo drastic conformational changes and one cannot infer their potential position without experimental data. For nsNSV, the structures during replication and transcription are still missing and therefore nobody knows the conformation during activity. More advanced analysis on sNSV polymerases (Influenza, Lassa,

SFTSV, La Crosse polymerases) show drastic conformational changes during replication and transcription. It is therefore very likely that large conformational rearrangement also occur for nsNSV L. The authors here go too far, their model does not have the necessarily basis and is therefore very likely to be incorrect. It would thus ask to remove this model and I would strongly recommend to state why the structure of an entire dimer cannot be inferred.

Response: Thanks for your comment! We agree with you that the CD-MTase-CTD position cannot be inferred currently although the CD-MTase-CTD (at least) are able to undergo drastic conformational changes. We have deleted supplementary Fig. 10b and stated why the structure of an entire dimer cannot be inferred as you suggested. However, based on our captured structure, we can infer that this dimer should not be symmetric, because if the two L-P copies are the same, serious steric clashes would be introduced between two L-P monomers from the modeling performed by aligning the second CD domain. The asymmetry of the hPIV3 L-P polymerase dimer is one of the similarities to the asymmetric influenza polymerase dimer. As a figure of symmetric dimer modeling can help authors understand the conclusion why the captured dimer should be asymmetric, we would like to retain this Fig. S10a as Fig. S13 in the revised manuscript.

5: the discussion on the replicase-encapsidase model for nsNSV (lines 249-262) is speculative. As said on point 4, Supplementary Figure 10 is not a structure, it is a speculative model that is not supported by experiment. In Supplementary Figure 11, the authors go further and propose that the replicase-encapsidase of nsNSV is similar to the one of Influenza. However, it is not known if the mechanism of encapsidation is similar for nsNSV. Indeed in Influenza, the 5' end of the nascent RNA that is newly synthesized by the replicase binds specifically in a conserved binding pocket of the encapsidase. This pocket has not been identified for nsNSV. This is acknowledged by the authors who put a question mark in their Supplementary Fig. 11 for the 5' of the nascent RNA. This is a central point of the replicase-encapsidase mechanism in Influenza. Therefore one cannot infer a model for non-segmented NSV L without structural data of an entire dimer. I would therefore recommend to be very careful in this discussion. I would recommend basing it on real data, i.e. analysing the charges on L+CD' as in Fig. 7e.

Response: Thanks for your comments! As you suggested, in the revised manuscript we are focused more on the positively charged surface of CD', and toned down the discussion about encapsidation and further highlight that the detailed mechanism of encapsidation in nsNSVs is still poorly understood. We fully agree with you that our study is the starting of the atomic structural and functional analysis of nsNSV L-P polymerase dimers and so the functional model that is presented is somewhat speculative at this stage. For this reason, we think it is appropriate to retain it as a supplement figure, with a clear statement in the text that it is a speculative model (lines 294–305, 316–319 and Fig. S14 legend).

6: line 264: the data does not reveal the entire L-L dimer interface. It reveals the interface between L and the CD' domain. This needs to be corrected.

Response: Thanks for your comment! The reprocessing result of class 1 particles revealed that the RdRp-PRNTase core of the second L should not contact the L copy 1, although its accurate orientation could not be confirmed. It is less likely that some other regions of L copy 2 contact L copy 1 in our captured L-P polymerase “dimer”, because if one region binds to the reconstructed “rigid” entire L, some additional densities bound to this L should be observed (usually protein-protein interfaces are relatively stable with well-resolved density). Therefore, our reported interface here should be the representative of the entire L-

L dimer interface, at least at this state we captured. Our 2D classification of potential entire dimers also showed that the bridging part between two L cores are a small region in most classes. In addition, the 2452 Å² total buried interface between RdRp-PRNTase and CD' of the second L is a very large interface. A comparable case is that of the reported asymmetric FluPol dimer (Carrique, L. et al. *Nature* 587, 638–643 (2020)) which has a total of about 2,590 Å² buried dimer interface including the host cofactor ANP32A. To be more careful, we have toned down the related description and added some explanation in the revised manuscript (lines 235–241).

In a nutshell, the data that is based on experiment is carefully analysed and meaningful. But the structure is not hPIV3 L dimer, it is hPIV3 L monomer+CD'. Discussion about the function of a dimer that is not structurally determined is therefore highly speculative.

Minor point:

1: line 62: two bands are seen for P in Fig 1b. It would be interesting to know what are these bands. Would it be possible to analyse them by mass spectrometry to see if it is a truncation or a phosphorylation?

Response: Thanks for your comment! We used mass spectrometry to detect the protein samples, but the quality of the obtained data was weak, and the deconvolution results showed random peak pattern without an identifiable molecular weight of P. This may be caused by complicated post-translational modifications of P protein during expression. We tried to optimize the experimental conditions, but unfortunately were unsuccessful.

2: lines 66-67: it would be interesting to know the percentage of identity between hPIV3 L, PIV5 L and NDV L. This is particularly important also because for NDV L as the conformation of hPIV3 L-P is similar to the one of NDV L-P.

Response: Thanks for your suggestion! The corresponding sequence identities have been added in this sentence (lines 71–72).

3: lines 101-103: it would be interesting to overlay (or show on two neighboring panels) hPIV3 RdRp active site and ebola RdRp bound to single-stranded template RNA to illustrate the sentence written.

Response: Thanks for your suggestion! We added a supplementary Fig. S5 to overlay hPIV3 RdRp active site and EBOV RdRp bound to single-stranded template RNA. In addition, RSV L-P structure bound to the template strand RNA was also published recently, so we also added this structure into Fig. S5.

4: lines 139-140: it would be interesting to add a figure that shows why the short beta-sheet formed with the beta-strand latch should be susceptible to disruption at elongation.

Response: We have added a supplementary Fig. 9 as you suggested.

5: lines 145-146: stabilization of the C-terminal region/core interaction through latch/lariat/beta-hairpin strut is also observed for segmented NSV polymerases (La Crosse, SFTSV and Lassa notably). This could be added as it seems to be important for stabilization of the CTER of very different NSV L.

Response: Thanks for your instructive suggestion! We have added this information in the revised manuscript (lines 175–178).

6: lines 152-156. This is difficult to follow because the residues mentioned in the text are not always visible on the related figure. This should be corrected.

Response: Corrected.

REVIEWERS' COMMENTS

Reviewer #2 (Remarks to the Author):

The authors have adequately addressed my concerns.

Reviewer #3 (Remarks to the Author):

The authors have carefully taken into account the most important points raised by the reviewers. I have no further objection to the publication of this important work.